# A compendium of chromatin contact maps reflecting regulation by chromatin remodelers in budding yeast

Hyelim Jo [1], Taemook Kim [1], Yujin Chun[1], Inkyung Jung [1] & Daeyoup Lee [1✉]

We herein employ in situ Hi-C with an auxin-inducible degron (AID) system to examine the effect of chromatin remodeling on 3D genome organization in yeast. Eight selected ATP-dependent chromatin remodelers representing various subfamilies contribute to 3D genome organization differently. Among the studied remodelers, the temporary depletions of Chd1p, Swr1p, and Sth1p (a catalytic subunit of the Remodeling the Structure of Chromatin [RSC] complex) cause the most significant defects in intra-chromosomal contacts, and the regulatory roles of these three remodelers in 3D genome organization differ depending on the chromosomal context and cell cycle stage. Furthermore, even though Chd1p and Isw1p are known to share functional similarities/redundancies, their depletions lead to distinct effects on 3D structures. The RSC and cohesin complexes also differentially modulate 3D genome organization within chromosome arm regions, whereas RSC appears to support the function of cohesin in centromeric clustering at $G_2$ phase. Our work suggests that the ATP-dependent chromatin remodelers control the 3D genome organization of yeast through their chromatin-remodeling activities.

[1] Department of Biological Sciences, Korea Advanced Institute of Science and Technology, 291 Daehak-ro, Yuseong-gu, Daejeon 34141, Republic of Korea.
✉email: daeyoup@kaist.ac.kr

Eukaryotic DNA is present in the nucleus in a highly organized form that is embodied through hierarchical folding steps. DNA containing genetic information is wrapped around a histone octamer to form a nucleosome, which acts as the basic structural unit of one-dimensional (1D) genome organization[1,2]. The nucleosome landscape is formed and maintained by the actions of specialized ATP-dependent chromatin remodelers, which have the functions of nucleosome sliding, spacing, assembly, eviction, and histone replacement. In *Saccharomyces cerevisiae* (*S. cerevisiae*), the ATP-dependent chromatin remodelers are divided on the basis of shared domains and functional similarities into four subfamilies: the chromodomain helicase DNA-binding (CHD), imitation switch (ISWI), INO80, and switch/sucrose non-fermentable (SWI/SNF) families[3–7]. For decades, researchers have studied the differential effects of these remodelers on the nucleosome. The CHD and ISWI subfamilies are well known to participate in nucleosome assembly and spacing[8–12]. The INO80 subfamily, which contains Ino80p, Swr1p and Fun30p in yeast, is mainly involved in histone variant exchange[13,14]. Finally, the SWI/SNF subfamily includes Sth1p and Snf2p, which are ATPase components of the Remodel the Structure of Chromatin (RSC) complex and the SWI/SNF complex, respectively, and act to modulate chromatin structure by nucleosome repositioning/ejection and histone eviction[15–17]. Since the nucleosome is incorporated in the 3D genome organization as a basic material of chromatin and the ATP-dependent chromatin remodelers can control nucleosome structure, we hypothesized that there could be a connection between ATP-dependent chromatin remodelers and 3D genome organization.

In *S. cerevisiae*, the systematic 3D genome organization of a characteristic Rabl configuration is characterized by clustering of centromeres, tethering of telomeres to the nuclear envelope, and sequestration of the ribosomal DNA (rDNA) locus[18–20]. Both chromosome arms extend from a centromere cluster as a pinning axis[18,21]. The 3D genome organization of yeast is not static; rather, it is dynamically controlled as the cell cycle progresses[22–24]. Chromosomes are gradually compacted as the intra-chromosomal interactions increase while the cell cycle progresses through interphase[22]. In contrast, the centromere clusters gradually loosen before the cell enters metaphase and then become denser as mitosis progresses[22,25].

Several previous studies suggested that the cohesin complex plays a pivotal role in higher-order genome organization and modulates the Rabl configuration in a CTCF-independent manner in *S. cerevisiae*[22,23,25,26]. The residency of cohesin at cohesin-associated regions (CARs) was reported to be closely related to 3D loop patterns in yeast[26]. Various ATP-dependent chromatin remodelers, such as Chd1p and the RSC complex, are known to interact with cohesin complexes[27–30]. In particular, the RSC complex contributes to the association and loading of cohesin complex on chromatin in the centromere and chromosomal arm regions[27,29–32]. Therefore, defects in the RSC complex impair sister chromatid cohesion and centromere structuring[29,30,33]. These correlations between the two complexes suggest that chromatin-remodeling mechanisms may actively participate in higher-order genome organization.

Here, we map the chromatin contacts that are affected by chromatin remodelers in budding yeast to elucidate the correlation between chromatin-remodeling activities and 3D genome organization. Our data show that each chromatin remodeler exhibits distinct activities relative to 3D genome organization, regardless of its subfamily membership or functional redundancy. Among the studied ATP-dependent chromatin remodelers, Chd1p, Swr1p, and Sth1p exhibit the strongest 3D genome-organizing activities. Our data further show that these three ATP-dependent chromatin remodelers play diverse roles according to the cell cycle stage and chromosomal context and, along with Scc1p, regulate centromere clustering at $G_2$ phase. In sum, we propose that chromatin-remodeling activity can directly modulate the 3D genome organization in yeast.

## Results

**ATP-dependent chromatin remodelers can affect 3D genome organization**. To focus on the actual function of ATP-dependent chromatin remodelers, we used an AID system to temporarily deplete target proteins[34]. In the presence of IAA (auxin, indole-3-acetic acid), an AID-tag-conjugated target protein is rapidly degraded by the artificially expressed E3 ligase, osTIR1[34]. In the present work, we targeted the ATPase subunits of yeast ATP-dependent chromatin remodelers representing various subfamilies, namely those encoded by *CHD1*, *SWR1*, *STH1*, *SNF2*, *INO80*, *ISW1*, *ISW2*, and *FUN30*[35]. After each chromatin remodeler was completely depleted, we performed in situ Hi–C to investigate how these depletions affected 3D genome organization[36]. To eliminate any bias arising from differences in sequencing depth, we normalized the in situ Hi–C dataset by using a random sampling method based on the minimal value of valid pair-reads (Supplementary Table 3).

As expected, the 3D genome structure was disorganized upon the temporary depletion of each ATP-dependent chromatin remodeler (Supplementary Fig. 1). Among the eight studied ATP-dependent chromatin remodelers, Chd1p, Swr1p, and Sth1p appeared to have the most dramatic activities in 3D genomic organization (Supplementary Fig. 1a–c).

Under the Chd1p- or Swr1p-depleted conditions, overall intra-chromosomal interactions increased. Analysis of the contact probability along genomic distance confirmed that there was an increase in short-to-intermediate (10–100-kb) distances under these conditions (Supplementary Fig. 1a–b, i–j). Ino80p depletion also increased the intra-chromosomal interaction, but the change was very weak compared to those seen under depletion of Chd1p or Swr1p (Supplementary Fig. 1e, m). Interestingly, although members of the ISW family are known to interact with Chd1p on chromatin, depletion of Isw1p or Isw2p appeared to have little effect on 3D genome organization (Supplementary Fig. 1f–g, n–o), as did deficiency of Fun30p (Supplementary Fig. 1h, p). The temporary depletion of Sth1p (the ATPase subunit of the RSC complex) caused an increase of intra-chromosomal interactions at intermediated distances (Supplementary Fig. 1c, k). In contrast, depletion of Snf2p, which is another member of SWI/SNF subfamily, had little effect on 3D genome organization (Supplementary Fig. 1d, l).

Our results suggest that each of the studied ATP-dependent chromatin remodelers plays a distinct role in 3D genome organization, regardless of its subfamily or homology. Among them, depletion of Chd1p, Swr1, and Sth1p yielded noticeable changes in 3D genome organization.

**The chromatin remodelers, Chd1p, Swr1p, and Sth1p, modulate 3D genome organization in a cell cycle-dependent manner**. As it is well known that 3D genome organization is dynamically controlled according to the cell cycle in yeast[22,37], we further investigated synchronized cells to dissect the functions of Chd1p, Swr1p, and Sth1p at various points along the cell cycle. Once cells were arrested at specific points in the cell cycle, each target protein was degraded by IAA treatment (Supplementary Figs. 2a and 3). Since the target proteins were depleted after cell cycle synchronization, further cell cycle progression was not affected by the loss of the target protein's activity (Supplementary Fig. 2b). Cells were harvested and alterations in 3D genome organization were quantified by in situ Hi–C.

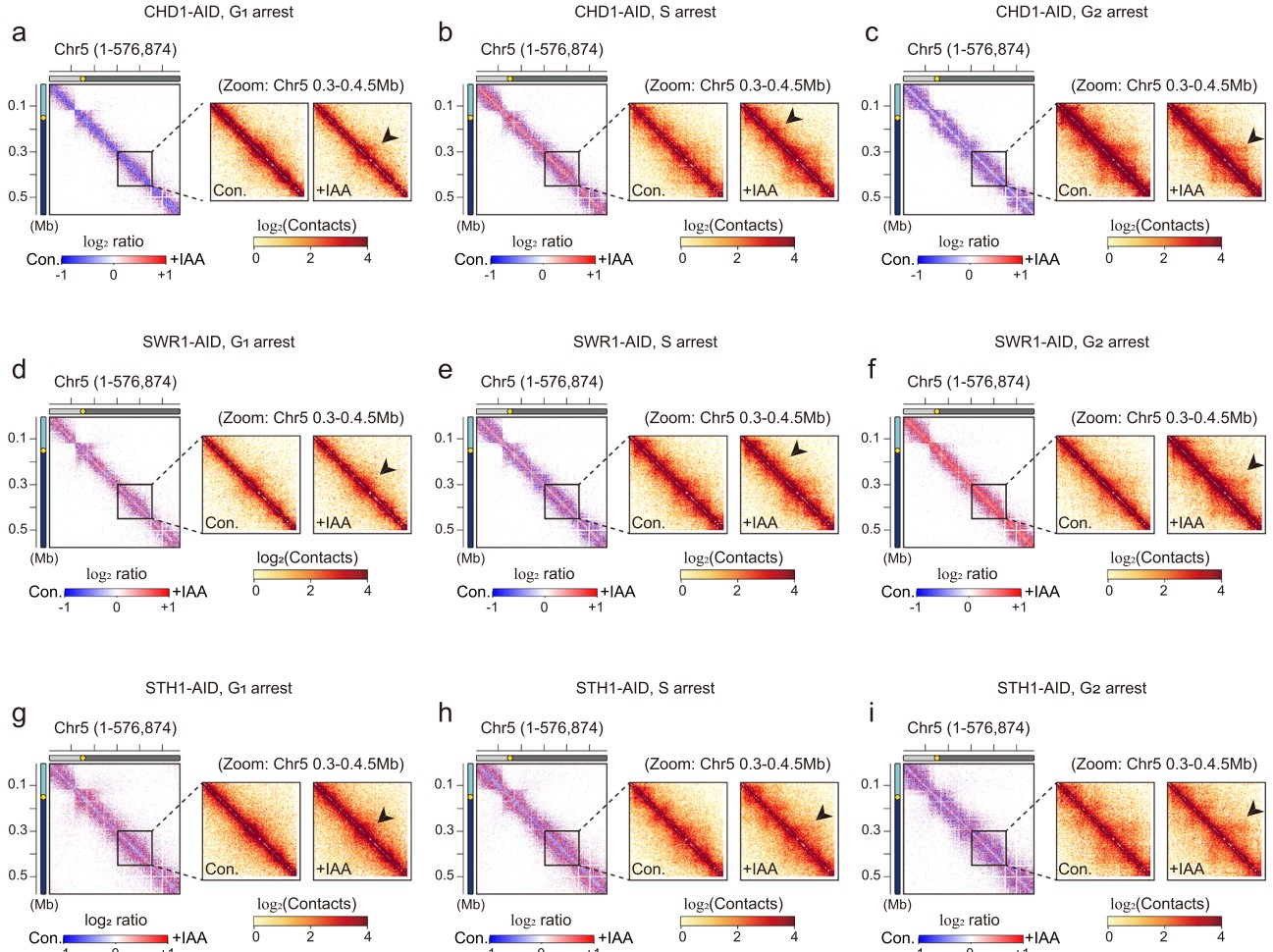

**Fig. 1 The 3D architecture of yeast chromatin is dynamically regulated by chromatin remodelers throughout the cell cycle. a–i** Contact maps (1-kb resolution) of chromosome 5 (576,874 bp) for CHD1-AID (a–c), SWR1-AID (d–f), and STH1-AID (g–i) strains at $G_1$ (a, d, g), S (b, e, h), and $G_2$ (c, f, i) phases. The left panels show the $\log_2$ ratio matrix of contact maps for knockdown versus control samples, and the two right panels show zoom-in matrices of chromosome 5 (0.3–0.45 Mb) for the control ('Con.') and knockdown ('+IAA') conditions. Schematic representations including genomic distances are displayed on the left side of each contact map. Yellow dots indicate the point centromere of the chromosome.

Our results revealed that Chd1p, Swr1p, and Sth1p showed surprisingly different effects depending on the cell cycle stage (Supplementary Fig. 4a–c). To investigate differences in intra-chromosomal contacts upon IAA treatment in more detail, we zoomed in on chromosome 5 (~576 kb), which is mid-sized among the *S. cerevisiae* chromosomes (Fig. 1a–i). Under the Chd1p-depleted condition, short-range intra-chromosomal contacts were collapsed at $G_1$ phase but increased in S phase, compared to control cells (Fig. 1a, b). In $G_2$ phase, Chd1p depletion had no significant difference relative to control on 3D genome organization, compared to those seen at $G_1$ or S phase (Fig. 1c). Under the Swr1p-depleted condition, overall intra-chromosomal interactions were distinctly strengthened compared to the control condition at $G_2$ phase, whereas little to no change was observed at $G_1$ and S phases (Fig. 1d–f and Supplementary Fig. 4d). Under the Sth1p-depletion condition, intra-chromosomal interactions were slightly decreased at very short distances and increased at intermediate distances for $G_1$ and S phases (Fig. 1g, h and Supplementary Fig. 4e). Sth1 had a more marginal effect on 3D genome organization in $G_2$ phase compared to $G_1$ and S phases (Fig. 1i).

The temporary depletion or permanent deletion of Sth1p is known to cause $G_2$ arrest[31,35,38]. Here, the $G_2$ cell accumulation initially noted under the Sth1p-depleted condition in asynchronous state was diminished by cell cycle synchronization: More marginal differences were observed in synchronous cells compared to asynchronous cells (Supplementary Figs. 1c and 4c).

Together, our results show that the ATP-dependent chromatin remodelers Chd1p, Swr1p, and Sth1p exhibit different impacts on 3D genome organization at different phases of the cell cycle.

**Despite being functionally redundant, Chd1p and Isw1p distinctly control 3D genome organization.** As mentioned above, Chd1p exhibited the most cell cycle-related difference in how its depletion affected 3D genome structure (Fig. 2a–c). Intra-chromosomal interactions at distances shorter than 100 kb were decreased by Chd1p depletion at $G_1$ and $G_2$ phases (Fig. 2d), with the largest decrease (1.5-fold) seen in very short-distance interactions (1 ~ 2 kb) under the $G_1$ arrest condition and only a weak decrease (<1.2-fold) observed under the $G_2$ arrest condition (Fig. 2d). In contrast, Chd1p depletion caused intra-chromosomal interactions to increase at S phase (up to 1.3-fold; Fig. 2d). Therefore, Chd1p seems to play greater roles in $G_1$ and/or S phase than $G_2$ phase. On the contrary, the depletion of Isw1p, which is well known to interact with Chd1p[11,12,39], had little effect on 3D genome structure even in $G_1$ phase (Fig. 2e, g and Supplementary Figs. 4a and 5a).

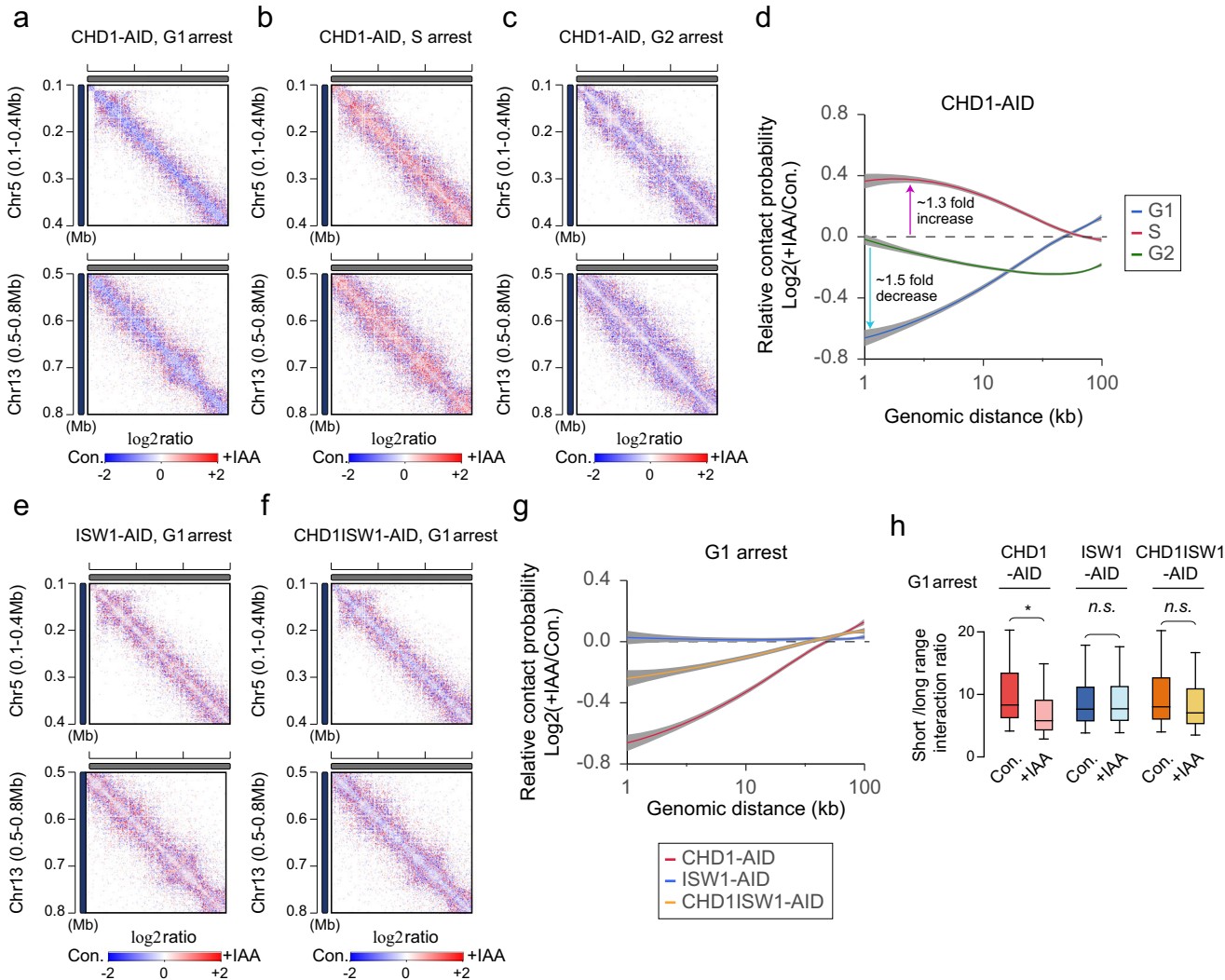

**Fig. 2 The function of Chd1p in 3D genome organization is distinct from that of Isw1. a–c** Zoom-in log$_2$ ratio contact map of chromosome 5 (0.1–0.4 Mb region; upper panel) and chromosome 13 (0.5–0.8 Mb region; lower panel) in CHD1-AID strain at G$_1$, S, and G$_2$ phases, respectively. The 1-kb resolution matrices of control and knockdown samples were used for log$_2$ ratio calculations. **d** Log$_2$ ratio of the average contact probability (CP) along genomic distance between control (Con.) and IAA-treated (+IAA) CHD1-AID strains at G$_1$, S, and G$_2$ phases. The gray shadow indicates the confidence interval around smooth (se). **e**, **f** Same as described for **a** but in ISW1-AID and CHD1ISW1-AID strains at G$_1$ phase. **g** Log$_2$ ratio of the average contact probability (CP) along genomic distance between control (Con.) and IAA-treated (+IAA) CHD1-AID, ISW1-AID, and CHD1ISW1-AID strains at G$_1$ phase. The gray shadow indicates the confidence interval around smooth. **h** Comparison of the short-versus-long range interaction (SVL) ratio per chromosome ($n = 16$) relative to 100 kb in CHD1-AID, ISW1-AID, and CHD1ISW1-AID strains at G$_1$ phase. The $p$-values were calculated using a one-sided Wilcoxon rank-sum test (*<0.05 and n.s. means not significant, the $p$-value of CHD1-AID; 0.041). Boxplot show median; box limits, upper and lower quartiles; whiskers.

To test whether defects in 3D genome structure caused by loss of Isw1p were hidden by its functional redundancy with Chd1p, we generated the double AID-tagging strain, CHD1ISW1-AID, which could simultaneously degrade both Chd1p and Isw1p. When both of these chromatin remodelers were depleted at G$_1$ phase, the contact map displayed a pattern intermediate between those generated by the individual depletions of Chd1p and Isw1p (Fig. 2f and Supplementary Fig. 5b). The relative contact probability curves also demonstrated that cells double-depleted of Chd1p and Isw1p yielded results that were intermediate between those obtained from cells depleted of Chd1p or Isw1p (Fig. 2g). Under the Chd1p-depleted condition, the short (<100 kb) versus long (>100 kb) interaction ratio (SVL) was also significantly decreased compare to that in control ('Con') cells; this was due to a reduction of short-to-mid-range interactions (Fig. 2g, h). These observations suggested that Chd1p could function to balance the SVL interaction ratio to maintain proper

chromatin conformation at G$_1$ phase. In contrast, the decrease of the SVL interaction ratio was not significant in the Chd1p and Isw1p double-depleted condition (Fig. 2h; compare 'CHD1-AID' with 'CHD1ISW1-AID'). As expected, the SVL interaction ratio was not altered under the Isw1p-depletion condition (Fig. 2h, 'ISW1-AID').

Taken together, our results indicate that Chd1p and Isw1p play distinct roles in 3D genome organization rather than being functionally redundant or similar. Furthermore, the synergistic malfunctions reportedly associated with double deletion of Chd1p and Isw1p in several prior studies[11,39] were not apparent in the context of 3D genome organization.

**The studied chromatin remodelers play distinct roles in 3D genome organization.** Since the ATP-dependent chromatin remodelers are globally distributed throughout the genome and

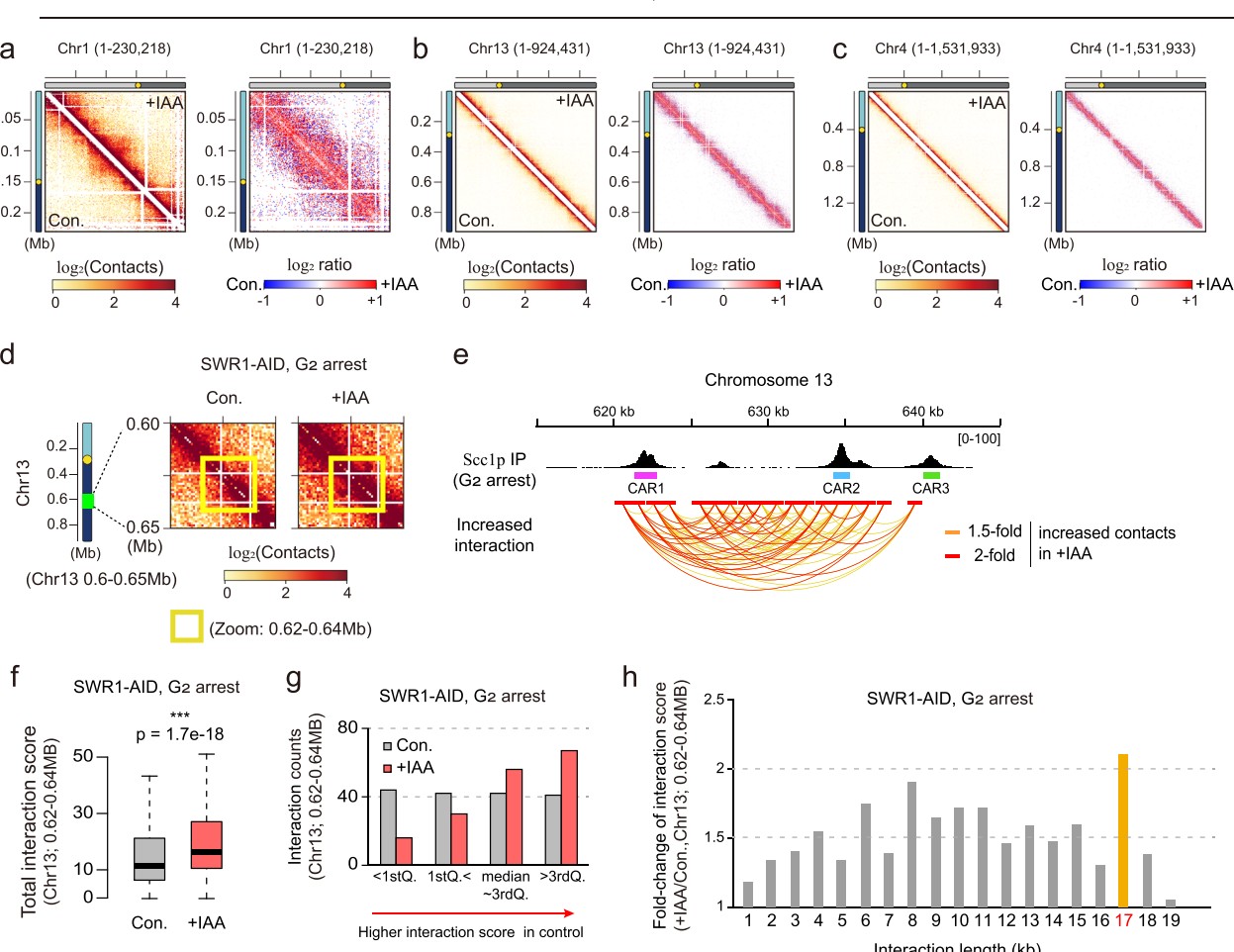

**Fig. 3 Swr1p modulates 3D genome organization in a manner that depends on the chromosomal context and/or cell cycle stage. a–c** Contact maps (1-kb resolution) of chromosome 1, 13, and 4, respectively, for SWR1-AID strain arrested at $G_2$ phase. The left panel shows the ICE-normalized matrix and the right panel shows the $\log_2$ ratio matrix of contact maps for knockdown versus control samples for each chromosome. **d** Zoom-in $\log_2$ ratio-interaction map of chromosome 13 (0.6–0.65 Mb region) for SWR1-AID strain at $G_2$ phase. Schematic representations including genomic distances and position are displayed on the left side of **d**. The yellow dots indicate the point centromere of the chromosome. The yellow box highlights a locus within 0.62–0.64 Mb. **e** (top) IGV data visualizing the localization of Scc1p on chromosome 13 under nocodazole-induced G2/M arrest. Pink, cyan, and yellow-green boxes indicate the Scc1p peak loci called the cohesin-associated region (CAR). Data deposited under accession number GSM4577764 was used for data analysis. (bottom) Arc plot displaying intra-chromosomal interactions within the 0.62–0.64 Mb region of chromosome 13. The red lines highlight contacts that showed >2-fold higher contact scores in +IAA samples compared to control samples, while orange lines highlight contacts that showed >1.5-fold higher contact scores in +IAA samples. **f** Boxplot comparing the interaction score of intra-chromosomal interactions (n = 170) within the 0.62-0.64 Mb region of chromosome 13 between control and +IAA samples. The p-value was calculated by a one-sided Wilcoxon rank-sum test. (***p-value < 0.001). Boxplot show median; box limits, upper and lower quartiles; whiskers. **g** The number of intra-chromosomal interactions within the 0.62–0.64 Mb region of chromosome 13 in control and +IAA samples. Total 168 contacts were divided into four groups (25%) by interaction score; (1) <1st Quantile, (2) >1st but <2nd Quantile (median), (3) >2nd but <3rd Quantile, and (4) >3rd Quantile. **h** The relative of intra-chromosomal interaction counts along the interaction distance within the 0.62–0.64 Mb region of chromosome 13.

display specific biochemical activities in chromatin-remodeling, we speculated that each ATP-dependent chromatin remodeler controls 3D contacts of the genome consistently across all chromosomes. Consistent with this hypothesis, we observed that depletion of a given ATP-dependent chromatin remodeler yielded the same disorganized pattern on chromosomes 1–4 (Supplementary Figs. 1 and 4). Swr1p, which appeared to largely involved in chromosomal decondensation at $G_2$ phase (Fig. 1f), also modulated the intra-chromosomal contacts of chromosomes 1, 4, and 13 in the same manner at $G_2$ phase (Fig. 3a–c). In *S. cerevisiae*, chromosome 1 is the smallest chromosome (~230 kb), chromosome 4 is the largest (~1,532 kb), and chromosome 13 falls between them in size (~924 kb). Thus, it seems

that the chromosome size does not have a huge effect on the ability of Swr1p to regulate 3D genome organization.

Depletion of Chd1p or Sth1p (Supplementary Fig. 6a, b) yielded similar results: The characteristic collapsed patterns seen upon Chd1p depletion at $G_1$ phase (an overall decrease of short-distance contacts) were equally evident on chromosomes 1, 4, and 13 (Supplementary Fig. 6a). Likewise, a decrease of very-short-distance contacts (~<10 kb) and an increase of intermediate contacts (~10–100 kb) were also observed on chromosomes 1, 4, and 13 under the Sth1p-depleted condition at $G_1$ phase (Supplementary Fig. 6b).

Collectively, these results show that most of the tested ATP-dependent chromatin remodelers (e.g., Chd1p, Swr1p, and Sth1p)

generally have a characteristic activity across all 16 chromosomes at a given phase of the cell cycle.

**Chromatin remodelers show diverse 3D genome-organizing functions depending on the chromosomal context at specific loci.** To investigate the 3D genome-organizing ability of the ATP-dependent chromatin remodelers in detail, we looked closely at their effects on individual chromosomes. When we zoomed in on the contact maps of individual chromosomes, we observed that the chromosomal-interacting domain (CID)- or loop-like positions were locally regulated in SWR1-AID strain. For instance, the intra-chromosomal contacts within a particular loop structure (dot-like shape in contact map) on chromosome 13 were strengthened (~1.5-fold) upon depletion of Swr1p at $G_2$ phase (Fig. 3d and Supplementary Fig. 7a), but not $G_1$ and/or S phase (Supplementary Fig. 7b, c). Thus, this specific regulation at ~0.62–0.64 Mb region appears to be cell cycle-stage specific. The depletion of Chd1p or Sth1p at $G_2$ phase also failed to induce a loop-like structure at this position (Supplementary Fig. 7d, e). We therefore conclude that this loop-like 3D structure at ~0.62–0.64 Mb on chromosome 13 is only manipulated and/or repressed by Swr1p at $G_2$ phase.

A previous study demonstrated that most loop positions are determined by CARs in yeast[26]. To investigate whether the Swr1p-modulated loop position was associated with CARs[26], we obtained previously reported data on the Scc1p peak position under nocodazole-induced $G_2$ arrest (Supplementary Fig. 7f)[26,40]. We then performed further experiments, which revealed that IAA treatment of SWR-AID strain increased the contact strength in numerous contacts between CARs (between CAR1, CAR2, and CAR3; Fig. 3e and supplementary Fig. 7g). Under IAA treatment, the average interaction strength in this region was significantly increased upon IAA treatment (Fig. 3f). The total interaction counts in this region were same but the interaction counts of the strong interactions (>3rdQ. group) were ~1.6-fold higher in the Swr1p-depleted condition (Fig. 3g and Supplementary Fig. 7h, i). The interaction counts of the weak interactions (<1stQ. group) were also decreased about 3-fold in the Swrp1-depleted condition (Fig. 3g and Supplementary Fig. 7i). Upon Swr1p depletion, contacts with distance of 17 kb mostly reflected increases in the interaction score of existing contacts (Fig. 3h). This increased contacts with distance of 17 kb mostly reflected the increased contacts between CAR1 and CAR3 (Fig. 3h and Supplementary Fig. 7j). A very recent report indicated that CAR-bound cohesins control the loop formation and extension[26]. Our data suggested that Swr1p depletion caused defects in this function of the CAR-bound cohesin. Indeed, we found that Swr1p was largely involved in modulating loop structures on all 16 chromosomes during $G_2$ phase (Supplementary Fig. 8a). The 266 loops detected across the 16 chromosomes exhibited increases or decreases upon IAA treatment of SWR1-AID strain at $G_2$ phase, but the average of strength of the loop signal was significantly increased after IAA treatment (Supplementary Fig. 8b, c). Among the detected loops, 92 increased >1.5-fold (Supplementary Fig. 8d), 32 decreased >1.5-fold, and the remaining 138 loops had marginal changes <1.5-fold upon IAA treatment. These findings suggest that Swr1p generally reduces the 3D contacts of chromosomes, but may also enhance 3D contacts within a specific local 3D structure-containing region.

Based on these results, we propose that each ATP-dependent chromatin remodeler can differentially modulate specific contacts of various chromatin regions by altering the chromatin architecture, in a manner that depends on the cell cycle and chromosomal context (e.g., the DNA sequence).

**The ATPase activity of Sth1p is necessary for its impacts on 3D genome organization.** Given our findings indicating that ATP-dependent chromatin remodelers can regulate 3D genome organization, along with the knowledge that these remodelers commonly share an ATPase domain that plays a pivotal role in their chromatin-remodeling activities, we next examined whether the chromatin-remodeling activities of the ATP-dependent chromatin remodelers are correlated with 3D genome-organizing processes in our system. Toward this end, we implanted a well-studied STH1K501R ATPase mutant into STH1-AID[27,41]. In the STH1-AID K501R mutant, normal Sth1p with an AID tag is degraded following IAA treatment, whereas Sth1p with a point mutation at K501 remains as an ATPase-inactive form.

Comparison of the patterns of change among 3D chromosomal contacts upon depletion of Sth1p with the K501R mutation revealed that the loss of Sth1p ATPase activity mimicked the 3D genome organization dysfunctions seen upon Sth1p depletion, even though the other domains of Sth1p remained (Fig. 4a, c and Supplementary Figs. 4c and 5c). More specifically, the following characteristic patterns were observed in both STH1-AID and STH1-AID K501R strains: First, Sth1p depletion decrease short-range interactions and increased mid-range interactions (Fig. 4a and Supplementary Fig. 4e), and a similar pattern was observed in STH1-AID K501R strain (Fig. 4c). Second, Sth1p depletion increased peri-centromeric interactions within a given chromosome at $G_1$ and S phase (Fig. 4a see black arrow and Supplementary Fig. 4c), and the same phenotype was observed in STH1-AID K501R strain (Fig. 4c see black arrow). Finally, the centromeric contacts at a distance of ~100 kb from the centromere were increased upon both Sth1p depletion and K501R mutation (Fig. 4b, d). These observations suggest that the chromatin-remodeling activity of Sth1p is necessary for its ability to modulate the 3D genome structure.

Next, we hypothesized that if there is a correlation between chromatin-remodeling and 3D genome organization, other enzymes whose depletion causes severe nucleosome structure alteration could also affect 3D genome organization. To test this hypothesis, we constructed AID strains for *SPT6*. Its encoded protein, Spt6p, is a highly conserved histone chaperone that is well known to be involved in eukaryotic transcription and to impact nucleosome occupancy[42–47]. As expected, Spt6p depletion also altered the genome-wide 3D chromosomal interaction at $G_1$ phase in a manner consistent with that seen following the depletion of the ATP-dependent chromatin remodelers tested herein (Supplementary Fig. 5d, f, g). Consistent with this result, a previous study revealed that depletion of another well-known chaperone, Spt16p, which is a subunit of FACT (facilitates chromatin transcription), also altered the 3D genome organization[31].

Together, these findings indicate that chromatin remodeling is connected to 3D genome organization, and that the ATPase activity of the ATP-dependent chromatin remodelers may directly regulate the 3D contacts of chromosomes in yeast.

**The function of Sth1p in 3D genome organization is largely distinct from that of Scc1p at $G_2$ phase.** Many studies over the decades have demonstrated that Sth1p cooperates with the cohesin complex. For example, Sth1p was found to physically interact with cohesin subunits and co-localize with the yeast cohesin loader in euchromatic regions[27,30,38], suggesting that Sth1p may modulate the loading of cohesin onto chromatin. Thus, we further investigated whether there could be a functional connection between Sth1p and cohesin in the context of 3D genome organization. To quantify the effects of Scc1p depletion

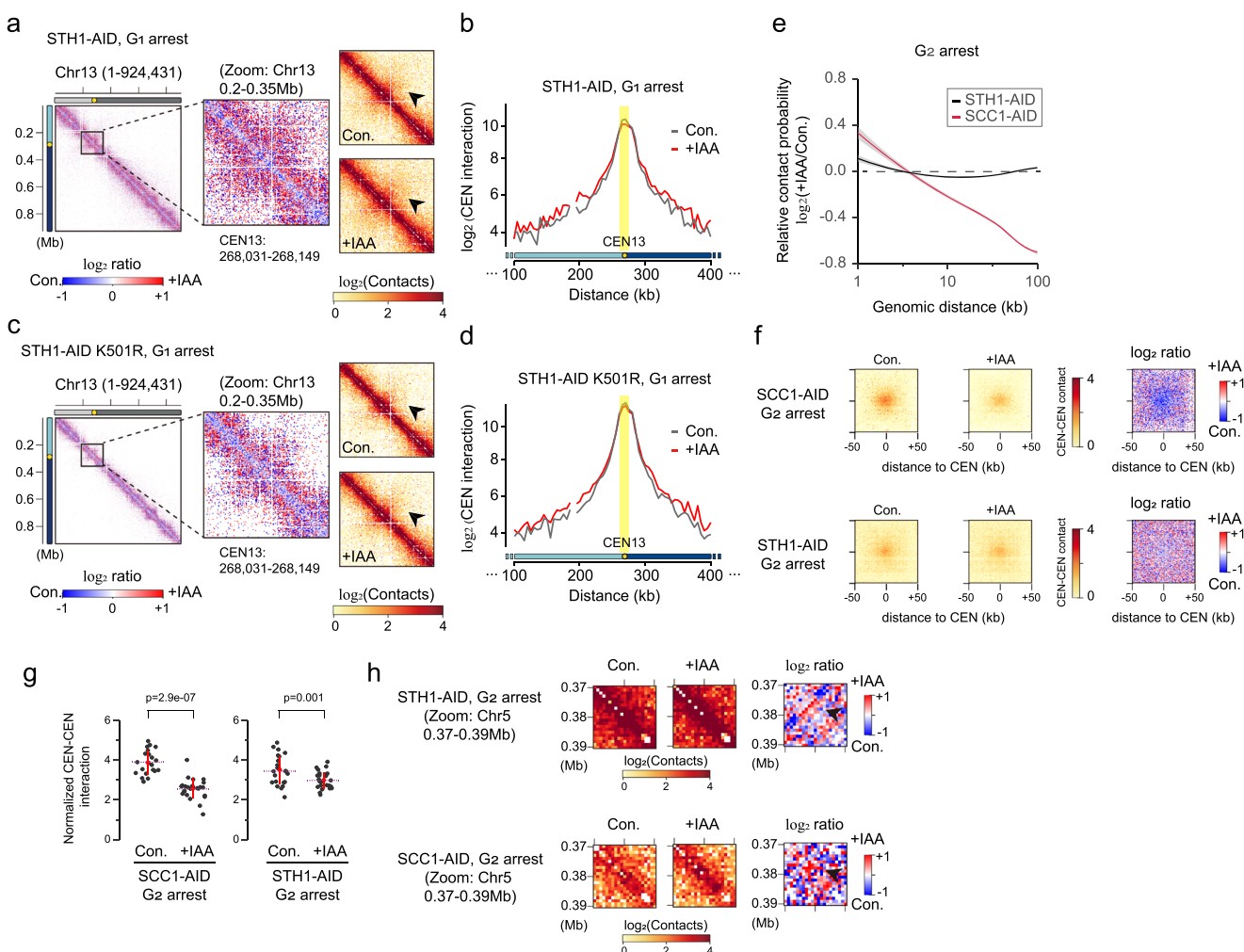

**Fig. 4 Sth1p has functions distinct from those of Scc1p in regulating 3D genome organization, except in the case of centromere clustering at $G_2$ phase.** **a** Contact map (1-kb resolution) of chromosome 13 for STH1-AID strain at $G_1$ phase. The black arrow indicates peri-centromeric interactions. **b** The distributions of intra-chromosomal contacts with the centromere locus on chromosome 13 in STH1-AID control (gray line) and +IAA (red line) samples at $G_1$ phase. The normalized contact value of the 5-kb-scale bin containing the centromere position was used for the calculation. The yellow dots indicate the point centromere of the chromosome. The centromere locus is highlighted in the yellow box. **c** Contact map (1-kb resolution) of chromosome 13 in STH1-AID K501R strain at $G_1$ phase. The black arrow indicates peri-centromeric interactions. **d** Same as described for **b**, but in the STH1-AID K501R control (gray line) and +IAA sample (red line). **e** Log₂ ratio of the average contact probability (CP) along genomic distance between control (Con.) and IAA-treated (+IAA) samples of SCC1-AID (red line) and STH1-AID (black line) strains at $G_2$ phase. The gray shadow shows display confidence interval around smooth. (Data deposited under accession number GSM2417297 was used for the SCC1-AID dataset). **f** Average matrices with 1-kb resolution showing inter CEN-CEN interactions (left panels) and their log2 ratios (right panels) in SCC1-AID (top) and STH1-AID (bottom) strains at $G_2$ phase. **g** The central 5 × 5 of the 1-kb scaled bins (n = 25) were used for plotting. The red line means standard deviation and pink dashed line represents the mean value. Statistical significance was measured using a one-sided Wilcoxon rank-sum test. **h** Zoom-in contact maps showing a narrow region (0.37–0.39 Mb) on chromosome 5 in STH1-AID (top) and SCC1-AID (bottom) strains at $G_2$ phase.

on 3D genome organization, we took advantage of previously published in situ Hi-C dataset of SCC1-AID strain at $G_2$ phase (GSM2417297)[22].

Given the reported functional correlation between the RSC and cohesin complex, we were surprised to observe that Scc1p had functions distinct from those of Sth1p in 3D genome organization: Unlike Sth1p depletion, which caused marginal changes throughout the cell cycle, the depletion of the yeast cohesin, Scc1p, caused overall disorganization of the 3D genome at $G_2$ phase (Fig. 4e). The relative contact probability at distances >10 kb was significantly diminished under depletion of Scc1p, but not Sth1p, at $G_2$ phase (Fig. 4e).

Interestingly, we found that Scc1p contributed to inter CEN-CEN interactions (Fig. 4f). A pile-up plot (an aggregate plot showing the contact strength between the centromeres of the 16 chromosomes) demonstrated that Scc1p depletion decreased the interactions of centromeric regions and their ±50 kb flanking regions (Fig. 4f upper panel). The inter CEN-CEN interactions also increased upon depletion of Sth1p, but this change was marginal compared to that seen upon Scc1p depletion (Fig. 4f bottom panel): The interaction strength between the central ±5-kb centromere-flanking regions was significantly reduced by about 1.6-fold under Scc1p depletion, whereas a relatively weak change of 1.2-fold was observed under Sth1p depletion (Fig. 4g).

These findings suggest that the cohesin, Scc1p, is sufficient to manage inter CEN-CEN interactions at $G_2$ phase, and that Sth1p may facilitate this function of Scc1p. Among the studied ATP-dependent chromatin remodelers, Chd1p and Swr1p were also

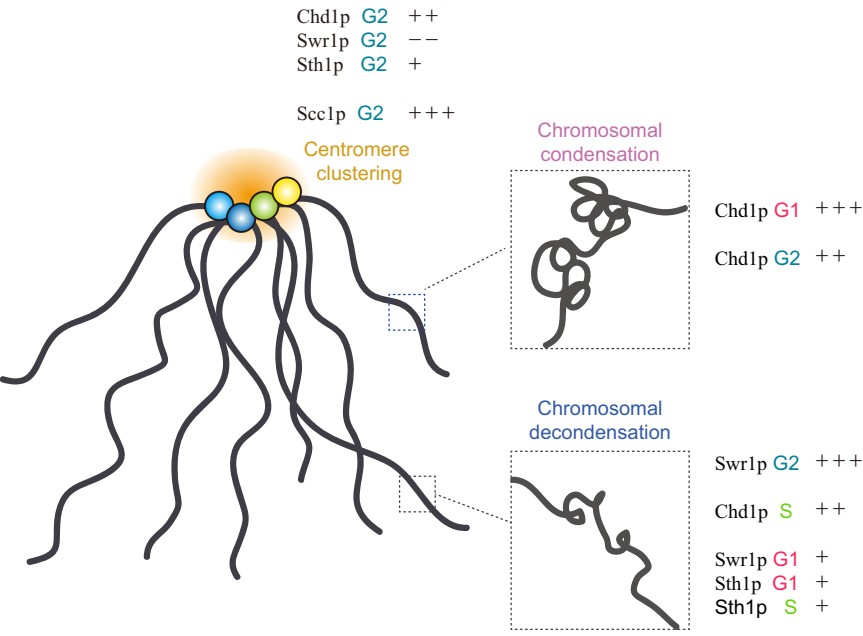

**Fig. 5 The regulatory functions of chromatin remodelers and cohesin in the Rabl configuration.** The schematic model summarizes the distinct roles of chromatin remodelers (Chd1p, Swr1p, and Sth1p) and cohesin (Scc1p) in 3D genome organization. We classified their roles based on the following three processes: (1) (intra) chromosomal condensation (pink); (2) (intra) chromosomal decondensation, (blue); and (3) centromere clustering (yellow). The degree of impact was expressed as a number of '+' (positive effect) or '−' (negative effect) symbols. Each cell cycle stage is marked with a different color ($G_1$, red; S, yellow green; $G_2$, teal blue).

found to contribute to inter CEN-CEN interactions at $G_2$ phase (Supplementary Fig. 9). Similar to Scc1p, Chd1p seemed to induce centromere clustering at $G_2$ phase, whereas Swr1p seemed to be involved in the loosening of centromere clustering at this time (Supplementary Fig. 9a, b).

The evidences for correlation between Sth1p and Scc1p on 3D genome organization were also detected on other loci. For example, the 0.37–0.39 Mb region of chromosome 5 was found to contain a locus that is commonly modulated by Sth1 and Scc1p (Fig. 4h). Examination of the log2 ratio map revealed that the bundle of chromosomal contacts near the 0.38-Mb locus increased under depletion of Sth1p or Scc1p, although the shapes of the increasing patterns differed between the two conditions (Fig. 4h right panels). This suggests that Sth1p and Scc1p may co-localize at specific regions, such as the 0.38-Mb locus on chromosome 5. Collectively, these results indicate that Sth1p can act with cohesin at certain loci and for certain functions (e.g., centromere clustering), whereas the two proteins have distinct functions at most chromosomal arm regions.

**Discussion**

Here, we conceptualized the comprehensive role of chromatin remodeling in 3D genome organization by performing in situ Hi-C experiments with AID strains in which we were able to temporarily degrade ATP-dependent chromatin remodelers. Our results demonstrate that the tested chromatin remodelers and the histone chaperone, Spt6p, all have differential effects on 3D genome organization, further suggesting that there are links between nucleosome structure and 3D genome organization.

Among the ATP-dependent chromatin remodelers, Chd1p, Swr1p, and Sth1p were found to differentially affect chromosomal contacts depending on the cell cycle stage and chromosomal context, as follows: (1) Chd1p contributed to chromosomal condensation at $G_1$ and $G_2$ phase, whereas it played a role in chromosomal decondensation at S phase. (2) Swr1p mostly participated in

chromosomal decondensation and regulation of loop structure in CARs, particularly at $G_2$ phase. (3) Sth1p was also involved in chromosomal decondensation, but primarily through mid-range chromosomal contacts (Fig. 5).

The cohesin complex plays a crucial role in 3D genome organization and is also known to affect nucleosome structures in a manner suggesting that there is a reciprocal relationship between nucleosome structure and 3D genome organization[31,32]. However, we herein found that the nucleosome structure in most euchromatic regions did not directly determine the 3D organizational pattern. A previous study suggested that the regularity of nucleosome spacing, but not the local nucleosome density, contributes to 3D genome-organizing mechanisms, such as local compaction[48]. This implies that chromatin-remodeling activities are more important than the nucleosomal landscape in determining the chromosomal 3D configuration.

Interestingly, we found that Sth1p and Scc1p modulated the contacts of chromosomal arms in different manners. This suggests that the mechanism of 3D genome organization cannot be fully interpreted by considering the collaborative RSC-cohesin complex pathway. Indeed, Sth1p showed diverse 3D genome-organizing functions depending on the chromosomal context and/or cell cycle stage. It implies that RSC can interact with several types of non-cohesin proteins depending on the cell cycle stage during 3D genome organization. We speculate that the extra functions of Sth1p exerted in collaboration with non-cohesin proteins are mainly involved in chromosomal arm decondensation.

At the centromere locus, in contrast to our findings in most euchromatic regions, there was a weak correlation between nucleosome structure and 3D centromere clustering. For example, both Sth1p and Scc1p were observed to play roles in centromere clustering. This suggests that the 3D structure of the centromere is organized independently of the euchromatin regions. The centromeres are pinned at a single point under the Rabl configuration, and are thus structurally separated from areas

where chromosome arms are crowded[18,19]. This separation of territory facilitates the cell's ability to strictly manage the 3D structure using a small number of architectural proteins. Here, we further show that chromatin remodeling can specifically modulate the 3D genome organization at confined locations.

Our 3D genome analysis of mutants in ATP-dependent chromatin remodelers demonstrated that nucleosome 3D genome architectures can be altered by some (but not all) chromatin remodelers throughout cell cycle progression. The chromatin remodelers exhibit distinct activities on 3D genome organization regardless of its subfamily membership or functional redundancy, even though all chromatin remodelers share common biochemical activities (e.g., the ability to alter histone-DNA interactions). Since we did not observe a consistent correlation between the nucleosome structure and 3D intra-chromosomal interactions in the in situ Hi–C work presented herein, more detailed studies using high-resolution technologies functioning at the gene-scale level, such as MicroC-XL, will be needed to resolve those relationships in detail.

In conclusion, our data demonstrated that the ATP-dependent chromatin remodelers can modulate chromosomal 3D configuration via their chromatin-remodeling activity depending on the chromosomal context and cell cycle stage.

## Methods

The yeast strains and primers used in this study are listed in Supplementary Tables 1 and 2. The valid pair-reads of in situ Hi–C are shown in Supplementary Table 3. The reproducibility of in situ Hi–C data are shown in Supplementary Table 4.

**Yeast strain generation**. In SPT6-AID and CHD1ISW1-AID strain, AID tagging of endogenous genes were performed by polymerase chain reaction (PCR) products with pKan-AID*-9myc plasmid. pKan-AID*-9myc was a gift from Helle Ulrich (Addgene plasmid # 99522; RRID:Addgene_99522). Endogenous BAR1 deletion was also performed by PCR. The STH1 gene cassette was PK (GKPIPNPLLGLDST) tagged and amplified with XmaI_Sth1_pro_F and bamHI_Sth1_PK_R primer set. The STH1K501R mutation was introduced by site directed mutagenesis PCR and was integrated into the Leu2 locus.

**Yeast cell harvest and cell cycle arrest**. In all experiments, pre-incubation (3 h) was performed to allow yeast to efficiently enter the mid-log phase. Thereafter, each yeast culture was re-diluted to 0.2 O.D.$_{600}$ and IAA (Sigma, I2886) was added to a final concentration of 0.5 mM for degradation of target proteins. An IAA stock (500 mM) was prepared in ethanol, and the same volume of 100% ethanol was used as a control.

For $G_1$ arrest, alpha-factor was added at a final concentration of 50 ng/ml to bar1Δ strains; after 2 h, 0.5 mM IAA was added and the cells were incubated for an additional 3 h. For S or $G_2$/M arrest, alpha-factor was added at a final concentration of 50 ng/ml to bar1Δ strains; after 1.5 h, the yeast cells were transferred to fresh YPD medium containing 200 mM hydroxyurea (HU; Sigma, H8627) for S arrest or 15 μg/ml nocodazole (Sigma, M1404) for $G_2$ arrest. At 1.5 h after HU/nocodazole treatment, 0.5 mM IAA was added and cells were incubated for an additional 3 h. For $G_2$ arrest, an additional 10 μg/ml of nocodazole was applied along with the IAA.

**Protein degradation test in AID strains**. The synchronized yeast cells were harvested after IAA treatment and lysed. The whole-cell extracts were suspended in SDS sample buffer and subjected to western blotting. For the AID strains, we utilized anti-Flag (Sigma F7425, 1:5000) and anti-Myc (Cell Signaling 2276S, 1:3000) antibodies. Anti-tubulin (Abcam ab6061, 1:5000) was used for a loading control. Anti-Rabbit (Jackson ImmunoResearch 111-035-003, 1:20,000), anti-Mouse (Jackson ImmunoResearch 115-035-003, 1:20,000), and anti-Rat (Jackson ImmunoResearch 112-035-003, 1:20,000) were used for secondary antibodies for anti-Flag, anti-Myc, and anti-tubulin, respectively.

**FACS**. The yeast DNA content was measured by FACS as described by Rosebrock et al. with some modifications[49]. Yeast cells were grown to an O.D.$_{600}$ of 1.0 and harvested, and the culture medium was removed. The cells were fixed by resuspension in 1 ml of 70% ethanol and stored overnight at −20 °C for at least 16 h. After fixation, resuspend cells with 500 μl of 50 mM Na-Citrate (pH7) and incubation for 10 min at room temperature. After rehydration step twice, cells were resuspended with 500 μl of 50 mM Na-Citrate (pH7) and stained using 2.5 μM Sytox Green (Invitrogen S7020) with 20 μg/ml RNaseA for 1 h at 37 °C. After staining, each sample was incubated with 10 μl of proteinase K (NEB, P8107S) for 1 h at 37 °C. The cells were sonicated (30%, 1 s ON/ 1 s OFF) and stored at 4 °C.

The cells were sonicated again and then a cytometric assay was performed using a BD LSRFortessa cell analyzer (BDbiosciences). The.fcs data were manipulated with the FCS Express software (De Novo Software).

**In situ Hi–C library preparation**. In situ Hi–C library preparation was performed as previously reported[50], with some modification of the steps designed to isolate yeast nuclei. Yeast cells (50 O.D.$_{600}$) were fixed with 3% formaldehyde (Wako, 064-00406) for 15 min and then quenched with 125 mM glycine. Quenched cells were pelleted and pre-incubated with β-ME buffer (20 mM EDTA and 0.7 M β-ME) for 10 min at 30 °C, and then lysed with 2 mg of zymolyase (US Biological, Z1004) in 2 ml lyticase buffer (1 M sorbitol, 50 mM Tris-Cl (pH 8.0), 5 mM β-ME) for 20 min at 30 °C. The obtained spheroplasts were resuspended in 2 ml of ice-cold PBS and 6 μg of pelleted nuclei were used for Hi–C library construction. The pelleted nuclei were resuspended in 50 μl of 0.5% SDS, incubated for 10 min at 62 °C and then immediately quenched with 170 μl of 1.47% TritonX-100 for 15 min at 37 °C. After lysis step, chromatin DNA was digested with 100 U of MboI (NEB, R0147) in 25 μl of 1XNEBuffer2 (NEB, B7002S) at least 2 h at 37 °C and subsequently incubated at 62 °C for 20 min to inactivate the MboI. To fill in the overhangs of restriction fragments and mark the DNA ends with biotin, each sample was incubated with 50 μl of fill-in master Mix: 37.5 μl of 0.4 mM biotin-14-dCTP (Life Technologies, 19524-016) and 1.5 μl of 10 mM dATP (Invitrogen, 18252015), dGTP (Invitrogen, 18254011), dTTP (Invitrogen, 18255018), and 8 μl (40 U) of Klenow fragment (NEB, M0210L) at 23 °C for 1.5 h with 500 rpm rotation. We then performed ligation with 2000 U of T4 DNA ligase (NEB, M0202L) at 23 °C for 4 h with slow rotation (300 rpm). After ligation, each sample was pelleted and resuspended with 550 μl of 1X TRIS buffer and then chromatin was decrosslinked overnight with 50 μl of 20 mg/ml of proteinase K (NEB, P8107S), 57 μl of 10% SDS and final 250 mM concentration of NaCl at 68 °C. DNA was purified using AMPure XP beads (Beckman Coulter) and sheared to 300–500 bp using a focused ultrasonicator (Covaris S220). After DNA shearing, fragments in the range of 200–600 bp were obtained using AMPure XP beads (Beckman Coulter). The biotinylated DNA was selectively purified using Dynabeads MyOne Streptavidin T1 beads (Life Technologies, 65601) and subsequently proceed to Hi–C library preparation using TruSeq DNA PCR-Free Low Throughput Library Prep Kit (Illumina, 20015962). The Hi–C library was quantified using a KAPA library quantification kit (Roche, KK4854) and further PCR amplification was performed using Phusion Hot Start II DNA polymerase (Thermo Fisher Scientific, F549S). The generated libraries were sequenced using 150-bp paired-end reads on an Illumina Novaseq6000 and/or HiSeqX.

**In situ Hi–C analysis**. The in situ Hi–C dataset was analyzed using a HiC-Pro pipeline[51] and sparse matrices were plotted using HiCPlotter[52]. The SaCcer3 S. cerevisiae genome was used as a reference genome. The contact maps of individual chromosomes were generated based on 1-kb resolution matrices, and other heatmaps containing the chromosomal contacts within chromosome 1(I) to chromosome 5(IV) were generated based on 5-kb resolution matrices. All of the processed matrices were normalized by the ICE method[53]. Random sampling was performed using the minimum value of valid pair-reads (Supplementary Tables 2 and 3). The contact probability plot was calculated according to the genomic distance and visualized using ggplot2 in R.

HiCcompare[54] was used to convert the 1-kb resolution of ICE-normalized matrices to a bedpe file, which was visualized using IGV[55] and quantified using ggplot2.

Loop detection and quantification analysis were performed using Chromosight[56] with the small-loop option and min_dist value = 100 kb.

The pile-up heatmaps for inter CEN-CEN interaction were generated base on 1-kb ICE-normalized matrices, given the small size of centromeres. The 1-kb scale bin that covered the point centromere was defined as a 'centromere bin', and the centromeric sub-matrix was extracted with ±50-kb flanking regions extending from each centromere bin. The centromeric sub-matrices that contained only inter CEN-CEN interactions were averaged and plotted using HiCPlotter[52].

The jitter plots for inter CEN-CEN interactions were generated with the values obtained for the central $5 \times 5$ (total 25) bins of the centromeric sub-matrix, using ggplot2. The $p$-value was calculated using a one-sided Wilcoxon rank-sum test.

The short-to-long distance interaction ratio was calculated with HiCExplorer[57,58] using the hicPlotSVL command.

The Pearson correlation coefficient between replicates was calculated by using R, and the stratum-adjusted correlation coefficient (SCC) for reproducibility of Hi–C data was calculated by HiCRep.py[59].

The in situ Hi–C data for SCC1-AID at $G_2$ stage was obtained from GEO, accession number GSM2417297[22].

**ChIP-seq analysis**. The ChIP-seq reads were mapped onto the sacCer3 reference genome using Bowtie2[60], and peak identification and downstream data analysis were performed using HOMER[61]. The ChIP-seq data for Scc1p were obtained from GEO, accession numbers [GSM2831174][40] and GSM4577764[26].

**Reporting summary**. Further information on research design is available in the Nature Research Reporting Summary linked to this article.

## Data availability

The data generated in this study are available from the corresponding author upon reasonable request. The in situ Hi–C data generated in this study have been deposited in the Gene Expression Omnibus (GEO) repository under accession code GSE158336. The processed in situ HiC data (cool format) are available at GEO database. The public datasets used in this study are available in the GEO database under accession codes: GSM2417297, GSM2831174, and GSM4577764. Source data are provided with this paper.

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

## Acknowledgements

The authors thank Nir Friedman and Helle D. Ulrich for gifts of AID strains. This work was supported by a National Research Foundation (NRF) of Korea Grant funded by the Ministry of Science and ICT (MSIT) (2018R1A5A1024261, SRC), and the Collaborative Genome Program for Fostering New Post-Genome Industry of the NRF funded by the MSIT (2018M3C9A6065070).

## Author contributions

H.J. and D.L. planned the study and H.J. performed experiments including in situ Hi–C library preparation and data analysis. H.J. and D.L. wrote the manuscript. T.K. assisted with the data analysis and Y.C. assisted with the in situ Hi–C experiments. I.J. advised on in situ Hi–C protocol.

## Competing interests

The authors declare no competing interests.
