## [Peer Review File · Nature Communications]

REVIEWER COMMENTS

Reviewer #1 (Remarks to the Author):

This work by Jo et al. explores the effect of chromatin modeling, and notably through the RSC complex, on the organization of the budding yeast genome. Depletion of the RSC catalytic subunit Sth1p prevents chromosome condensation and centromere clustering, in coordination or not with cohesin, respectively. The authors used the auxin-inducible degron AID to induce the depletion of ATPase subunits of eight ATP-dependent chromatin remodelers, and analyze the impact on chromosome folding using Hi-C. They identified Sth1p as having the largest effect. The paper therefore presents new results on a range of RSC complex molecules, and provides new insights on the regulation of chromosome folding. It is unfortunate that (if I understand correctly), all of these analyses were done on asynchronous cell populations (but for Sth1). This will limit the interest to reuse the data in other studies. I understand that the idea was to screen a broad range of mutant strains, but still, it would have been interesting, and not so much additional work, to do both G1 and G2/M arrest for each mutant immediately.

Concerns

- The results are somehow difficult to interpret, as the paper flows. For instance, in Figure 1, I have not seen indicated what are the growth condition of the cell used for the contact maps? The text is not mentioning it either. The methods section suggests these are asynchronous, but it is really unclear. The authors must explain clearly what the reader is looking at! And if we are looking at asynchronous cells contact maps in Figure 1, why is that? These populations contain a mix of G1, G2/M, etc. with very different structures. Therefore, it is not very informative to draw conclusions from such experiments. The authors should state clearly these limitations (again, if this is the case). In the final chapter of their result, they dissect the effect of the cell cycle on the Sth1 depletion. I think this chapter should follow immediately the first chapter, presumably done on asynchronous cells. It would be more logical to me to immediately address this point.
- The use of Pearson Correlation is absolutely not informative to assess the reproducibility between experiment. More suitable methods such as HiCRep exist, although the interest to compare genomewide maps is very limited (<https://pubmed.ncbi.nlm.nih.gov/28855260/>). It is more informative to compare the outcome of analysis on the datasets together, or to focus on subsamples of the maps. For instance, the authors could call loops and compare the results in the different maps they are looking at (using for instance programs such as Chromosight to quantify the loops strengths; <https://www.biorxiv.org/content/10.1101/2020.03.08.981910v3>)
- Calling loops would clearly be an interesting addition to the work, especially in contact maps from cell populations arrested in G2/M. Even in asynchronous cells, looking at contact enrichment between known sites of cohesin deposition should give a good insight on the proper deposition of the loops in these mutants. Besides doing ratio plots of their contact maps, the authors should do a better job at performing quantitative analysis of the features present in them.
- Chromosomal context: the authors look at three specific chromosomes which always display outlier behaviors in Hi-C. Chr III carries the mating type loci, is super small, is full of LTR that complicate reads alignment, and harbor a specific folded structure due to the mat organization. Chr. VI is super small. Chr. XII carries of course the rDNA. It is fine to discuss these observations, but the authors must also present a couple of "standard" chromosomes!
- The analysis concerning the clustering of centromeres is interesting; the same could be done for telomeres?
- What about the transcription status of the cells? The authors could run a RNA-seq analysis (or reuse

published dataset if available), and discuss the outcome in light of folding modifications.

- Several of the contact maps ratios are quite noisy (as shown in Supplementary Figure 3). The random sampling normalization procedure used may result in the loss of information.

To improve their maps and analysis, the authors should try alternative methods using recently developed signal strength-dependent binning, such as:

<https://www.ncbi.nlm.nih.gov/pmc/articles/PMC7320618/>

<https://www.ncbi.nlm.nih.gov/pmc/articles/PMC6486590/>

- One possible interpretation of the results presented in Figure 2 on the effect of Sth1p depletion is that the declustering of the centromeres impacts the steric constraints brought about by the "polymer brush" configuration in the peri-centromeric regions, by reducing the grafting density there. This physical effect must be considered when interpreting the data.

In addition, the authors should back their Hi-C analysis with an imaging analysis of centromere positions in the cell.

More information could be found here:

<https://link.springer.com/article/10.1186/s13059-017-1199-x>

- "In chromosome III, both proteins were observed to be involved in specific interactions in regions associated with the mating type, albeit to different degrees (Fig. 3e, g. black arrow)."

-> More quantitative analyses should support this assertion. A magnification with genomic annotations is needed, to visualize the different interactions and partners. In the present state, I don't see what the authors aim at showing in the Figure.

minor points

- The raw data should be made available, so that the referees can appreciate their quality.

- Concerning the impact of the cell cycle on the spatial organisation of yeast genomes, it would be relevant to also mention the work concerning the yeast *S. Pombe* (Tanizawa et al. Nat Struct Mol Biol 2017 Nov;24(11):965-976

- Please improve the legends of the color scale of the ratio maps to help the unfamiliar reader. Clearly indicate what are the conditions looked at, not just +/- IAA... (especially Fig.2)

- Please plot (in the sup figures or in the figures) the local derivative of the contact probability curves. This plot facilitates the reading of the curves by magnifying the differences.

- Bin sizes should be indicated in all figure legends.

- What is the GSE118214 dataset? It seems to be from Klein-Brill et al, please cite the paper when using the data.

Reviewer #2 (Remarks to the Author):

In this manuscript, Jo et al. assessed the impact of different chromatin remodelers on the 3D organization of the genome as monitored by HiC. Individual chromatin remodelers were inactivated by inducing the degradation of their respective ATPase subunits STH1, SNF2, CHD1, SWR1, INO80, ISW1, ISW2, and FUN30 using the auxin degron system.

Growth defects could be observed after depletion of the essential factors Sth1 (the ATPase subunit of the RSC complex) and Ino80 as expected.

Depleting the ATPase subunits has different impacts on chromosome conformation, Sth1depletion having the strongest impact. The main caveat of this study is that it is very difficult to estimate the direct versus indirect effects of depleting these factors. In particular, it is almost impossible to

interpret these data without knowing the impact of each of these depletions on the cell cycle. Many if not all of the defects observed could be just due to an accumulation of cells in one specific phase of the cell cycle; Ideally, all these experiments should have been performed in synchronized cells to rule out cell cycle effects. Actually, most of the effects observed in the Sth1 depleted cells are barely visible in synchronized cells (compare ratio maps in figure 1 vs figure 6 for Sth1). Other indirect effects could result from impaired transcription or replication. It is noteworthy that Sth1 leads to a global decrease in transcription as documented in (Parnell et al, 2008) for the three polymerases. Furthermore, the consequence of depleting each of these ATPases on nucleosome occupancy has been studied by the Friedman lab (Klein-Brill et al., 2019) using the same system. This paper is cited but its results and conclusions are not discussed to interpret the data on 3D organization obtained here. Likewise, Sth1 depletion has been studied, by several groups and most of this work is not cited here (Kubik et al, 2018; Hartley and Madhani, 2009; Parnell et al. 2008). It will be important to consider these data to interpret this work. Overall, it is very difficult to draw any conclusion from the data presented in this manuscript. The literature is poorly cited, some controls are missing and some data are miss interpreted (see specific examples below).

Figure 1: as mentioned above, these data can't be interpreted without knowing the impact of these depletions on the cell cycle. FACS profiles must be provided.

P3: "Chromosome XII, which harbors the rDNA locus, is condensed into a spherical shape and becomes localized to the nucleolus at interphase^{18,19,26}. The disorganization of a spherically condensed rDNA locus coincides with mitotic cell division ^{22,26}."

The literature is not properly cited here, none of these papers is describing the rDNA as a spherical shape.

The Gadal lab recently published a very detailed analysis of the structure of the rDNA over the cell cycle (ref 26) showing that "During interphase, rDNA was progressively reorganized from a zig-zag segmented line of small size (5,5 μm) to a long, homogeneous, line-like structure of 8,7 μm in metaphase"

Ref 19 is not about the rDNA...

P7: "These findings suggest that Sth1p participates in rDNA decondensation."

These observations could also be the consequence of the inactivity of the rDNA as it was shown that Sth1 depletion leads to a massive decrease of polI transcription (Parnell et al, 2008).

Figure 3: controls for spt6 depletion are missing. Western blot showing the effective depletion of the protein should be provided as well as FACS profile 3hrs after AID treatment.

Figure 5:

Figure 6:

Comparing panels 1a and 6b or 6g and 2k it is obvious that the effects of depleting Sth1 are lost or severely reduced when cells are synchronized. This strongly argues that most of these effects are due to the accumulation of cells in a specific phase of the cell cycle, probably G2/M.

In general, it is not very easy to appreciate small differences on ratio-maps, mainly because of the low signal to noise ratio on distant contacts (Baudry et al, 2020). The authors could try using Serpentine a recently developed algorithm that adapts the bin size to generate low noise ratio maps.

P12: "Interestingly, we found that Sth1p modulated the contacts of chromosomal arms in a manner

independent of Scc1p and Scc2p.” I am afraid that there is a major flaw in the reasoning here. This manuscript does not provide any evidence that Sth1 effects on chromosome organization are independent of Scc1 and Scc2. To draw such a conclusion, one would need to monitor the effect of Sth1 depletion in the absence of Scc1 or Scc2. Yet, here the authors compared the depletion of Sth1 with the one of Scc1 or Scc2 and observed opposite effects on chromosome “condensation”. This is exactly what one could expect if Sth1 depletion led to the accumulation of cells in G2/M, when chromosomes are condensed by cohesins. Sth1 depletion would lead to more condensation while Scc1 or Scc2 depletion would lead to less condensation.

Reviewer #3 (Remarks to the Author):

Differential regulation of 3D genome organization by chromatin remodeler RSC complex

The authors investigated the effect of remodeler loss of 3D genome organization using Hi-C in *S. cerevisiae*. It was found that loss of numerous remodelers/chaperones caused perturbations in inter- and intra-chromosome interactions with loss of Sth1 having an especially severe impact, which also appears to be cell-cycle dependent and mainly distinct from cohesin function. The work is of good quality, and the questions asked are interesting. There are a few comments/questions:

1. Because the manuscript relies heavily on heatmaps, a better introduction to interpreting Hi-C data would be helpful to those less familiar with the technique (e.g. an extra panel spelling out what each part of the heatmap represents). Some more information in the text about the normalization method would also be welcome.

2. Related to the above, it is not always obvious what is being referred to as an effect caused by remodeler loss (e.g. in Fig 1c, it is difficult to see what the three arrows are pointing to). Would it be possible to enlarge some portions of the plot (as with the CEN interactions) to better demonstrate what is alluded to in the text? This would mainly be beneficial for Fig. 1.

3. RSC is well-known to act on -1/+1 nucleosomes throughout the cell. Are gene transcription start sites overrepresented as sites of interaction?

4. Line 178-179: “A malfunction in the decondensation activity of Sth1p increases random contacts along the chromosomal arms” What is the rationale for assuming that the wt contacts are nonrandom and the mutant contacts are random? Is it possible that in mutant cells, favorable contact points are instead shifted vs wt cells rather than randomly distributed?

5. Is it possible that remodelers with some redundancy (e.g. Isw1 & Isw2) have a greater impact on 3D genome organization that is overlooked here? This is especially relevant as it is noted in the discussion that regular nucleosome spacing is important for 3D genome organization.

6. I believe on line 357 “iced-normalized” should be “ICE-normalized”

Summary of the reviewers' major comments, and experiments performed to address them.

Let me begin by stating how much I appreciate the three reviewers who took the time to make critical comments on manuscript. We have carefully revised the manuscript to take into account the reviewers' major concerns.

The major points in our revised manuscript are followings;

Point 1. Cell cycle synchronization

As we agreed with the common major concerns, we performed additional work for *in situ* Hi-C on synchronous cells. We picked up *CHD1-AID*, *SWR1-AID* and *STH1-AID* strains which exhibited dramatic changes in asynchronous state. Also, we also used *ISW1-AID* strains (for negative control) which exhibited no changes in asynchronous state.

Point 2. Title

As we observed phenomena worth reporting in synchronous state, we expanded our focus from Sth1p (RSC complex) to Chd1p, Swr1p and Sth1p (Chromatin remodeler). Thus, we modified the title of our manuscript from '*Differential regulation of 3D genome organization by chromatin remodeler RSC complex*' to '*A compendium of chromatin contact maps regulated by chromatin remodelers in budding yeast*'.

Point 3. Resolution (Signal to noise)

To improve signal to noise, we used 1kb resolution rather than 5kb resolution. We updated most of data including calculation (e.g., contact probability) and visualization (e.g., heat-map) to using 1kb resolution matrices.

Point 4. Chromosome examples

To demonstrated more generous situation, we visualized the chromosome 1, 4, 5 and 13 rather than chromosome 3, 4 and 12. We chose the chromosome 1 for the smallest size, chromosome 4 for the biggest size and the chromosome 5, 13 for representing mid-range size chromosomes.

Point 5. Usage of magnified image

To communicate accurately the visualization data to our readers, we tried to show zoom-in contact maps rather than contacts maps of chromosome from 1 to 4.

** The following secure token has been created to allow review of record GSE158336.

Token: uxmfysgwdpatbkb

Point-by-point response to reviewer's comments

Reviewer #1 (Remarks to the Author):

This work by Jo et al. explores the effect of chromatin modeling, and notably through the RSC complex, on the organization of the budding yeast genome. Depletion of the RSC catalytic subunit Sth1p prevents chromosome condensation and centromere clustering, in coordination or not with cohesin, respectively. The authors used the auxin-inducible degron AID to induce the depletion of ATPase subunits of eight ATP-dependent chromatin remodelers, and analyze the impact on chromosome folding using Hi-C. They identified Sth1p as having the largest effect. The paper therefore presents new results on a range of RSC complex molecules, and provides new insights on the regulation of chromosome folding. It is unfortunate that (if I understand correctly), all of these analyses were done on asynchronous cell populations (but for Sth1). This will limit the interest to reuse the data in other studies. I understand that the idea was to screen a broad range of mutant strains, but still, it would have been interesting, and not so much additional work, to do both G1 and G2/M arrest for each mutant immediately.

We sincerely appreciate the reviewer's comments which have helped us to improve our manuscript. We completely agree with the reviewer's comments and have taken advantage of the reviewer's input by adding new experiments and analyses, thus substantially revising the manuscript.

Concerns

- The results are somehow difficult to interpret, as the paper flows. For instance, in Figure 1, I have not seen indicated what are the growth condition of the cell used for the contact maps? The text is not mentioning it either. The methods section suggests these are asynchronous, but it is really unclear. The authors must explain clearly what the reader is looking at! And if we are looking at asynchronous cells contact maps in Figure 1, why is that? These populations contain a mix of G1, G2/M, etc. with very different structures. Therefore, it is not very informative to draw conclusions from such experiments. The authors should state clearly these limitations (again, if this is the case). In the final chapter of their result, they dissect the effect of the cell cycle on the Sth1 depletion. I think this chapter should follow immediately the first chapter, presumably done on asynchronous cells. It would be more logical to me to immediately address this point.

Our comment : We took your advice and generated a brand-new Figure 1 using synchronous cells. In revised manuscript, we also tried to indicate each cell cycle state on all of the figures, respectively.

(Please refer to revised Fig. 1.)

- The use of Pearson Correlation is absolutely not informative to assess the reproducibility between experiment. More suitable methods such as HiCRep exist, although the interest to compare genomewide maps is very limited (<https://pubmed.ncbi.nlm.nih.gov/28855260/>). It is more informative to compare the outcome of analysis on the datasets together, or to focus on subsamples of the maps. For instance, the authors could call loops and compare the results in the different maps they are looking at (using for instance programs such as Chromosight to

quantify the loops strengths; <https://www.biorxiv.org/content/10.1101/2020.03.08.981910v3>)

Our comment : We agree with Pearson correlation is not proper method for assessing reproducibility on *in situ* Hi-C. We also tried to use the Euclidean distance instead of Pearson correlation because of this issue but there was no difference in sensitivity between two methods.

As you mentioned above, HiCRep was quite suitable for *in situ* Hi-C (*compared to Pearson correlation or Euclidean distance*) so we had a good experience with this tool. However, it was not sufficiently sensitive for focus on subsamples of the maps. For instance, in Supplementary Table 4, the SCC (Stratum adjusted correlation coefficient) values between two of different cell cycle state (e.g., G₁ vs G₂) were above 0.8.

Thus, we decide to display the SCC values alongside with Pearson correlation coefficient value in Supplementary Table 4.

(Please refer to revised Supplementary Table 4.)

- Calling loops would clearly be an interesting addition to the work, especially in contact maps from cell populations arrested in G₂/M. Even in asynchronous cells, looking at contact enrichment between known sites of cohesin deposition should give a good insight on the proper deposition of the loops in these mutants. Besides doing ratio plots of their contact maps, the authors should do a better job at performing quantitative analysis of the features present in them.

Our comment : Following the comment mentioned above, we used Chromosight to call the loop position. Chd1p and Sth1p seemed to have little effect on loop structures but Swr1p had an impact on loop structure in G₂ phase.

As you can see below images, Swr1p modulated the strength of loops at G₂ phase. It seemed to there were no new-generated loops upon Swr1p-depletion. Instead, Swr1p appeared to regulate only already existed loops.

However, reporting these findings has been controversial because of the following concern; Since the signal of the yeast chromatin loops is weaker in the interphase (even in G₂/M) compare to actual mitotic phase, the result of loop detection by Chromosight contains abundant non-loop positions in interphase. (We used default 'small loop' option same as

Matthey-Doret et al., 2020)

Thus, we exhibited only a sample image for SWR1-AID strains (Supplementary Fig. 5a).

(Please refer to revised Supplementary Fig. 5a.)

- Chromosomal context: the authors look at three specific chromosomes which always display outlier behaviors in Hi-C. Chr III carries the mating type loci, is super small, is full of LTR that complicate reads alignment, and harbor a specific folded structure due to the mat organization. Chr. VI is super small. Chr. XII carries of course the rDNA. It is fine to discuss these observations, but the authors must also present a couple of “standard” chromosomes!

Our comment :

Chr	Size	
chr4	1531933	: is the biggest one
chr15	1091291	
chr7	1090940	
chr12	1078177	: for rDNA locus
chr16	948066	
chr13	924431	
chr2	813184	
chr14	784333	
chr10	745751	
chr11	666816	
chr5	576874	
chr8	562643	
chr9	439888	
chr3	316620	: for mating type locus
chr6	270161	
chr1	230218	

For the reasons mentioned above, we have shown three chromosomes before. In response to the opinion that more standard examples are needed, we chose chromosome 1, 4, 5 and 13 rather than chromosome 3, 4, 12. The chromosome 1 represent smallest chromosome while chromosome 4 represent biggest chromosome. The chromosome 5 and 13 represent mid-size of standard chromosome.

Thus, we have revised the manuscript in a way that shows the more generous and standard situation.

- The analysis concerning the clustering of centromeres is interesting; the same could be done for telomeres?

Our comment : Unfortunately, the contacts of telomere locus were only can observed in chromosome 1, 6 and 3 (small chromosomes under 400kb). We thought that the three of telomeres cannot represent all of telomere region so we did not proceed data analysis in this region.

Since this telomere issue did not improve with increasing of total read-out (5Gb to 10Gb), it might be needed other method such as a second crosslinker to enhance the contacts within telomere regions.

- What about the transcription status of the cells? The authors could run a RNA-seq analysis (or reuse published dataset if available), and discuss the outcome in light of folding modifications.

Our comment : We already have done RNA-seq analysis in asynchronous cells but we did not found a correlation with transcription and 3D genome organization in even Chd1p or Swr1p which were not affected by cell cycle population.

Since the average size of yeast genes is about 1.3kb, it would be supposed to the resolution of in situ Hi-C was not sufficient for metagene analysis. We suggest that the high-resolution methods such as 'microC-XL' were needed to study connection between transcription and 3D genome organization.

- Several of the contact maps ratios are quite noisy (as shown in Supplementary Figure 3). The random sampling normalization procedure used may result in the loss of information. To improve their maps and analysis, the authors should try alternative methods using recently developed signal strength-dependent binning, such as:
<https://www.ncbi.nlm.nih.gov/pmc/articles/PMC7320618/>
<https://www.ncbi.nlm.nih.gov/pmc/articles/PMC6486590/>

Our comment : To improve the signal-to-noise ratio, we used a 1kb resolution matrix instead of a 5kb resolution matrix. As a result, it can provide a clearer contact map to readers.

- One possible interpretation of the results presented in Figure 2 on the effect of Sth1p depletion is that the declustering of the centromeres impacts the steric constraints brought about by the "polymer brush" configuration in the peri-centromeric regions, by reducing the

grafting density there. This physical effect must be considered when interpreting the data. In addition, the authors should back their Hi-C analysis with an imaging analysis of centromere positions in the cell.

More information could be found here:

<https://link.springer.com/article/10.1186/s13059-017-1199-x>

Our comment : It is an interesting interpretation but these phenotype about centromeric interaction was enhanced due to cell cycle accumulation upon Sth1-depletion in asynchronous cells.

We also did not consider imaging analysis because there were previous studies that Sth1p is involved in centromere clustering or sister chromatid segregation including microscopic data (e.g, Hsu et al., 2003; Prasad et al., 2019 but in *Candida albicans*)

In this work, we wrote a conceptualized a compendium of effects of chromatin remodelers on 3D chromatin contact maps. However, if we continue with further work related to the features observed in this work, we might even try to prove the mechanism by directly comparing high-resolution contact maps and high-resolution microscopy data.

- "In chromosome III, both proteins were observed to be involved in specific interactions in regions associated with the mating type, albeit to different degrees (Fig. 3e, g. black arrow)." -> More quantitative analyses should support this assertion. A magnification with genomic annotations is needed, to visualize the different interactions and partners. In the present state, I don't see what the authors aim at showing in the Figure.

Our comment : It is very good comment but the figures related with chromosome 3 were removed. In revised data, we tried to display the figures with quantitative analysis.

minor points

- The raw data should be made available, so that the referees can appreciate their quality.

Our comment : The reviewers can use this token for GSE158336 / uxfyfsgwdpatbkb

- Concerning the impact of the cell cycle on the spatial organisation of yeast genomes, it would be relevant to also mention the work concerning the yeast *S. Pombe* (Tanizawa et al. Nat Struct Mol Biol 2017 Nov;24(11):965-976

Our comment : If it is helpful to understanding 3D genome organization in yeast, we willing to cite Tanizawa et al., 2017. We updated the citation in introduction part.

- Please improve the legends of the color scale of the ratio maps to help the unfamiliar reader. Clearly indicate what are the conditions looked at, not just +/- IAA... (especially Fig.2)

Our comment : We have updated the scale bar in consideration of our readers.

- Please plot (in the sup figures or in the figures) the local derivative of the contact probability curves. This plot facilitates the reading of the curves by magnifying the differences.

Our comment : In this case, we zoomed in the curves instead of using the local derivative curves for an intuitive grasp.

- Bin sizes should be indicated in all figure legends.

Our comment : We indicated resolution in revised figure legends.

- What is the GSE118214 dataset? It seems to be from Klein-Brill et al, please cite the paper when using the data.

Our comment : We eliminated data related with GSE118214 but clearly cited the paper on other GSE data set.

Reviewer #2 (Remarks to the Author):

In this manuscript, Jo et al. assessed the impact of different chromatin remodelers on the 3D organization of the genome as monitored by HiC. Individual chromatin remodelers were inactivated by inducing the degradation of their respective ATPase subunits STH1, SNF2, CHD1, SWR1, INO80, ISW1, ISW2, and FUN30 using the auxin degron system. Growth defects could be observed after depletion of the essential factors Sth1 (the ATPase subunit of the RSC complex) and Ino80 as expected.

Depleting the ATPase subunits has different impacts on chromosome conformation, Sth1 depletion having the strongest impact. The main caveat of this study is that it is very difficult to estimate the direct versus indirect effects of depleting these factors. In particular, it is almost impossible to interpret these data without knowing the impact of each of these depletions on the cell cycle. Many if not all of the defects observed could be just due to an accumulation of cells in one specific phase of the cell cycle; Ideally, all these experiments should have been performed in synchronized cells to rule out cell cycle effects. Actually, most of the effects observed in the Sth1 depleted cells are barely visible in synchronized cells (compare ratio maps in figure 1 vs figure 6 for Sth1). Other indirect effects could result from impaired transcription or replication. It is noteworthy that Sth1 leads to a global decrease in transcription as documented in (Parnell et al, 2008) for the three polymerases. Furthermore, the consequence of depleting each of these ATPases on nucleosome occupancy has been studied by the Friedman lab (Klein-Brill et al., 2019) using the same system. This paper is cited but its results and conclusions are not discussed to interpret the data on 3D organization obtained here. Likewise, Sth1 depletion has been studied, by several groups and most of this work is not cited here (Kubik et al, 2018; Hartley and Madhani, 2009; Parnell et al. 2008). It will be important to consider these data to interpret this work. Overall, it is very difficult to draw

any conclusion from the data presented in this manuscript. The literature is poorly cited, some controls are missing and some data are miss interpreted (see specific examples below).

Our comment : First of all, we would like to thank for critical comments and also make it clear that we fully respect the results of previous studies and do not want to deny them.

The reason why we have minimized several citations and proper interpretation of previous studies, as mentioned in the discussion section, it was difficult to find the relationship between the 3D genomic organization and the existing transcription or 1D nucleosomal structure. Considering this situation, we thought that interpretation with existing studies should be approached carefully with sufficient evidences. Therefore, we wrote a reporting-type manuscript based on the facts we observed, excluding extravagant argument.

We apologize that if you felt that our opinion or conclusion was lacking and please review the revised manuscript through additional work.

Figure 1: as mentioned above, these data can't be interpreted without knowing the impact of these depletions on the cell cycle. FACS profiles must be provided.

Our comment : The FACS profiles provided in revised Supplementary Fig. 2.

P3: "Chromosome XII, which harbors the rDNA locus, is condensed into a spherical shape and becomes localized to the nucleolus at interphase^{18,19,26}. The disorganization of a spherically condensed rDNA locus coincides with mitotic cell division^{22,26}."

The literature is not properly cited here, none of these papers is describing the rDNA as a spherical shape.

The Gadai lab recently published a very detailed analysis of the structure of the rDNA over the cell cycle (ref 26) showing that "During interphase, rDNA was progressively reorganized from a zig-zag segmented line of small size (5,5 µm) to a long, homogeneous, line-like structure of 8,7 µm in metaphase"

Ref 19 is not about the rDNA...

Our comment : Good comment, but the rDNA-related contents have been removed. We removed the data on chromosomes 3 and 12 because we accepted feedback asking for more standard situations.

P7: "These findings suggest that Sth1p participates in rDNA decondensation."

These observations could also be the consequence of the inactivity of the rDNA as it was shown that Sth1 depletion leads to a massive decrease of poll transcription (Parnell et al, 2008).

Our comment : As mentioned above, rDNA related contests have been removed from the revised manuscript. If we continue further research about rDNA locus, we are willing to investigate the aspects of the transcription and RNA Pol II.

Figure 3: controls for spt6 depletion are missing. Western blot showing the effective depletion of the protein should be provided as well as FACS profile 3hrs after AID treatment.

Our comment : As the supplementary data section was fully composed of asynchronous data set, the western blotting data was excluded in supplementary figure (*In fact, these Western data were removed preferentially as they overlap with the data from Friedman's lab for chromatin remodelers.*). Spt6 was also well degraded by IAA treatment (*Please see below original test experiment*).

The FACS profiles of *SPT6-AID* provided in revised Supplementary Fig. 2.

Figure 5

Figure 6:

Comparing panels 1a and 6b or 6g and 2k it is obvious that the effects of depleting Sth1 are lost or severely reduced when cells are synchronized. This strongly argues that most of these effects are due to the accumulation of cells in a specific phase of the cell cycle, probably G2/M.

Our comment : As we accepted the concern about cell cycle, we performed additional works with synchronous cells for accurate interpretation the phenomenon occurred in *STH1-AID* strains. We also mentioned this issue on revised manuscript ('*The temporal depletion of Sth1p known to caused G₂ arrest as well as permanent deletion of this protein^{32,39,40}. Since these G₂ cell accumulation in Sth1p-depleted condition was diminished after cell cycle synchronization, more marginal differences were observed at synchronous cells compared to those observed in asynchronous cells.*').

In revised manuscript, we described function of Sth1p only with synchronous state (it was weaker than asynchronous state though).

In general, it is not very easy to appreciate small differences on ratio-maps, mainly because of the low signal to noise ratio on distant contacts (Baudry et al, 2020). The authors could try using Serpentine a recently developed algorithm that adapts the bin size to generate low noise ratio maps.

Our comment : To improve signal to noise issue, we have tried to several methods including suggested tool. Finally, we decided to use 1kb resolution matrix instead of 5kb resolution matrix. It provides a high-quality contacts map and also no contact signal loss.

P12: “Interestingly, we found that Sth1p modulated the contacts of chromosomal arms in a manner independent of Scc1p and Scc2p.” I am afraid that there is a major flaw in the reasoning here. This manuscript does not provide any evidence that Sth1 effects on chromosome organization are independent of Scc1 and Scc2. To draw such a conclusion, one would need to monitor the effect of Sth1 depletion in the absence of Scc1 or Scc2. Yet, here the authors compared the depletion of Sth1 with the one of Scc1 or Scc2 and observed opposite effects on chromosome “condensation”. This is exactly what one could expect if Sth1 depletion led to the accumulation of cells in G₂/M, when chromosomes are condensed by cohesins. Sth1 depletion would lead to more condensation while Scc1 or Scc2 depletion would lead to less condensation.

Our comment : We carefully selected words from the revised manuscript. To interpret the relationship between Sth1p and Scc1p, we avoided expressing it as 'independent'.

However, we still have not found similar change patterns between *STH1-AID* and *SCC1-AID* in G₂ phase so it is clear that they have distinct role at G₂ phase. Also, Sth1p exhibited only marginal effects on 3D structure of chromosomes throughout the cell cycle while cohesin mainly involved in 3D genome organization. Altogether, we concluded that they have somewhat distinct activities in case of the 3D genome organization.

Instead, we found that there were tiny loci where were affected by both of Sth1p and Scc1p. It implies that they are co-localized at this locus. Thus, in consistent with previous research (Munoz et al., 2019), Sth1p seems to participate in the cohesin (and/or loader) localization at this locus in G₂ phase.

Reviewer #3 (Remarks to the Author):

Differential regulation of 3D genome organization by chromatin remodeler RSC complex
The authors investigated the effect of remodeler loss of 3D genome organization using Hi-C in *S. cerevisiae*. It was found that loss of numerous remodelers/chaperones caused perturbations in inter- and intra-chromosome interactions with loss of Sth1 having an especially severe impact, which also appears to be cell-cycle dependent and mainly distinct from cohesin function. The work is of good quality, and the questions asked are interesting.

Our comment : We would like to thank you for your interest in our work.

There are a few comments/questions:

1. Because the manuscript relies heavily on heatmaps, a better introduction to interpreting Hi-C data would be helpful to those less familiar with the technique (e.g. an extra panel spelling out what each part of the heatmap represents). Some more information in the text about the normalization method would also be welcome.

Our comment : To accommodate a broad reader in a wide field of biology, we tried to revise the text and figure legends kindly. The figure legends were separated as much as possible, and the scale bar also modified to help readers understand more intuitively. In the normalization method, we cited original paper which contain professional explanation.

2. Related to the above, it is not always obvious what is being referred to as an effect caused by remodeler loss (e.g. in Fig 1c, it is difficult to see what the three arrows are pointing to). Would it be possible to enlarge some portions of the plot (as with the CEN interactions) to better demonstrate what is alluded to in the text? This would mainly be beneficial for Fig. 1.

Our comment : In response to this comment, most of the main figures were modified with enlarged plots or matrices.

3. RSC is well-known to act on -1/+1 nucleosomes throughout the cell. Are gene transcription start sites overrepresented as sites of interaction?

Our comment : Since the yeast genes are very small (average 1.3kb), the resolution of *in situ* Hi-C is not sufficient for metagene analysis. We suggest that the high-resolution methods such as 'microC-XL' were needed to study connection between transcription (and also -1/+1 nucleosome) and 3D genome organization.

4. Line 178-179: "A malfunction in the decondensation activity of Sth1p increases random contacts along the chromosomal arms" What is the rationale for assuming that the wt contacts are nonrandom and the mutant contacts are random? Is it possible that in mutant cells, favorable contact points are instead shifted vs wt cells rather than randomly distributed?

Our comment : In fact, 'random' may not have been an appropriate word to interpret this phenomenon. We wanted to express the chaotic state that the rules for chromosomal contact were disappeared.

It can be assumed that there is a certain position that is favorably remodeled by normal chromatin remodeler in wild type cells. This position probably provides a 3D contact point for cohesin complex. If this contact point is covered by nucleosome (in mutants), however, the 3D contact may appear other position instead of this covered position. If it is occurred by misspacing of nucleosome, it would be shifted within 100bp distance.

However, the Sth1p exhibited overall decrease of short-range contacts (under 10kb) and increase of mid-range contacts. It means that the chromosomes show the preference for mid-range contacts comparing to short-range contact in the Sth1p-depletion condition. Thus, we chose the 'random' rather than 'shift' in this case.

It is an interesting part to discuss but the lines 178-179 were deleted from the revised manuscript because we reconstructed all the main figures. If someday we can track an individual 3D contact in a novel method, we can finally get the answer to this question.

.

5. Is it possible that remodelers with some redundancy (e.g. Isw1 & Isw2) have a greater

impact on 3D genome organization that is overlooked here? This is especially relevant as it is noted in the discussion that regular nucleosome spacing is important for 3D genome organization.

Our comment : In response to point 5, additional work was done using the *Isw1p* and *Chd1p* double knockdown strain. We chose *Isw1p* and *Chd1p* because they had the most serious synergistic effects on the nucleosomal structure in previous studies (Ocampo *et al.*, 2016).

Interestingly, there was no greater impact on 3D genome organization upon double knock-down. It means that the functional redundancy between CHD and ISWI families did not appear in 3D genome organization. Instead, they appear to play a distinct and different role in 3D genomic organization. It was also proven again that the nucleosomal structure did not directly determine the 3D structure of the chromatin.

(Please see Fig.2)

6. I believe on line 357 “iced-normalized” should be “ICE-normalized”

Our comment : We are very sorry for confused description

Again, we thank the reviewers for their time and effort reviewing our manuscript. The comments were constructive and, in addressing them, we have greatly improved our insight into chromatin remodelers in 3D genome organization..

Sincerely,

Daeyoung Lee, PhD.

P.S. The following is a list of changes made to the revised manuscript:

- 1) To respond Reviewer #1's specific comment 1 and Reviewer #2's comment 5, we performed *in situ* Hi-C using the G₁, S and G₂ phase arrest cells in *CHD1-AID*, *SWR1-AID*, *ISW1-AID* and *SPT6-AID*.
- 2) To respond Reviewer #3's specific comment 5, we examined whether there is a functional redundancy between *Chd1p* and *Isw1p* on 3D genome organization. We performed *in situ* Hi-C in *CHD1ISW1-AID* (double AID) at G₁ phase.
- 3) To respond Reviewer #1's specific comment 1, We have reorganized the main figures to include only the synchronous state.
- 4) As following changes in 3), the supplementary figures were also reorganized with the data of asynchronous state.

- 5) We also completely revised result section of manuscript to provide a proper interpreting reorganized main figures.
- 6) We changed the title of our manuscript to include the results of additional works.
- 7) We made the appropriate corrections on the abstract and discussion to include a result of additional work.
- 8) We additionally uploaded total 40 of data-set in SRA repository.
- 9) To respond Reviewer #2's specific comment 1 and 4, we provided FACS profiles after cell cycle arrest and compared the ethanol(control) and IAA treatment.
- 10) To respond Reviewer #1's specific comment 7 and Reviewer #2's comment 6, the signal to noise ratio on contact maps was improved by using 1kb resolution.
- 11) As following changes in 9), we completely revised analysis and calculation with 1kb resolution.
- 12) To respond Reviewer #1's specific comment 4, we visualized the chromosome 1, 4, 5 and 13 rather than chromosome 3, 4 and 12 to display more generous situation.
- 13) As following changes in 11), we zoomed in specific loci on the 'standard chromosome' rather than demonstrated the specialized loci (mating type or rDNA locus).
- 14) To respond Reviewer #3's specific comment 2, we used the magnified contact maps rather than the contacts maps of chromosome from 1 to 4
- 15) To respond Reviewer #1's minor comment 3, we modified the scale bar of contact maps.
- 16) To respond Reviewer #1's specific comment 1, we indicated the cell cycle stage in all figures.
- 17) To respond Reviewer #1's minor comment 5, we indicated the bin size in figure legends.
- 18) To respond Reviewer #3's comment 1, we divided main figure and revised the manuscript and figure legends to make it easier for a broader reader to understand.
- 19) To respond Reviewer #1's specific comment 2, we assessed the reproducibility of *in situ* Hi-C experiments by using HiCRep in Supplementary table 4.
- 20) To respond Reviewer #1's specific comment 2 and 3, we detected loop position by using Chromosight in Supplementary Fig. 5a.
- 21) To respond Reviewer #2's specific comment 7, we revised an expression which using a 'independent' to interpret the relationship between Sth1p and Scc1p.
- 22) To respond Reviewer #1's minor comment 2 and 6 and also Reviewer #2's specific comment 2, we modified citation.

REVIEWER COMMENTS

Reviewer #1 (Remarks to the Author):

The revised version of the manuscript by Jo et al. "A compendium of chromatin contact maps regulated by chromatin remodelers in budding yeast" addresses most of the concerns raised during the first round of revision.

I find the article much more complete and I thank the authors for having added multiple experiments and analysis. I still have concerns regarding the quality of the English, which remains hard to read and sometimes to understand (for instance, in the abstract: "The regulatory roles of chromatin remodelers on 3D genome organization were appeared to irrelevant with functional redundancy or similarity" is incorrect and unclear). If the authors had the opportunity to have the text reviewed by an English speaker, that would help. Alternatively, using a tool such as deepl.com could also greatly improve the clarity of the text.

The work is quite descriptive, as mentioned by the authors in their response to reviewer 2, and provide an exhaustive description of the effects of remodeling factors at a relatively large scale. The authors should mention that using finer techniques to map contact changes, such as microC, may bring up additional details at the gene-scale level.

I regret that the authors were not able to run a loop calling detection tool (chromosight or else, but that should be cited in the text) with more success, and I don't really understand their response to the original suggestion: "Since the signal of the yeast chromatin loops is weaker in the interphase (even in G2/M) compare to actual mitotic phase,..." Sup figure 5a is not really convincing, and I am sure that the analysis could have been done with much better results would the author have played with the program a bit. There are ways to quantify the loops strengths and changes over the entire genome, to show the differences (or not) between IAA treated or not treated condition for each mutant, etc. Here the analysis looks preliminary and superficial, and I am not sure it really brings much to the text.

The use of references is also sometimes a bit strange.

line 65: then become denser as mitosis progresses^{25,26}. Refs 25 and 26 refers to meiosis, not mitosis. The authors probably think about schalbetter et al. 2017, not 2019? Lazar et al 2017 also shows this. Same for the "in a CTCF-independent manner in *S. cerevisiae*^{22,23,26-28}": 26 should be schalbetter 2017, no? Ref 27 and 28 are in mammals, and should be cited earlier in the sentence, I think, and not after mentioning yeast... etc. The authors should double check their citations.

fastq files must be made available (and not just .cool files), so far they are not.

Reviewer #2 (Remarks to the Author):

This revised version of the manuscript addressed one of my main concerns, which was the difficulty of interpreting changes due to remodeling depletion in asynchronous populations. Data on synchronized cells are now presented and have confirmed the importance of comparing synchronized populations. Indeed, the new set of data shows that the effect observed for Sth1 was mainly due to the accumulation of cells in G2 upon depletion of Sth1.

FACS are now provided for the different conditions, which is nice, but Western blots to monitor the depletion are not shown.

The effects of depleting individual remodelers (assuming that the depletion was effective in each case) on synchronized populations are very small. This is not a problem per se but raises the question of the significance of these observations for which no statistics are provided. Furthermore, the description of the data is either vague (e.g. p5 "ino80 depletion also slightly increased interchromosomal contacts") or exaggerated (e.g. p 5 "among them depletion of Chd1, Swr1 and Sth1p yielded dramatic changes in 3D genome organization" or p10 "These observations imply that chromatin remodeling activity

plays a key role in modulation of 3D genome structure » when the effects are barely visible and never quantified).

In response to my comment about the poor signal-to-noise ratio (also raised by reviewer 1), the authors choose to show interaction matrices at a 1kb resolution which is not helping at all to solve this issue. Matrices are even more noisy and difficult to interpret.

None of the analysis proposed by reviewer 1 and 2 was applied in this revised version and none of the observation is validated by a different approach.

In general, this new version is poorly written and is sometimes even difficult to understand.

The raw data may be of interest to the community and these results could be published as a resource but not as a research article, in the current state.

Reviewer #3 (Remarks to the Author):

The authors have adequately addressed our comments in the revision. We feel the manuscript can be accepted.

Summary of the reviewers' major comments and the experiments performed to address these comments

We sincerely thank the reviewers for providing detailed and insightful comments and suggestions. These comments and suggestions have helped us greatly improve the manuscript. We revised the manuscript and added quantitative analyses. We also worked to improve the English writing and avoid vague expressions.

Please find below our detailed response to the reviewers' comments.

Point-by-point response to reviewer's comments

Reviewer #1 (Remarks to the Author):

The revised version of the manuscript by Jo et al. "A compendium of chromatin contact maps regulated by chromatin remodelers in budding yeast" addresses most of the concerns raised during the first round of revision.

I find the article much more complete and I thank the authors for having added multiple experiments and analysis. I still have concerns regarding the quality of the English, which remains hard to read and sometimes to understand (for instance, in the abstract: "The regulatory roles of chromatin remodelers on 3D genome organization were appeared to irrelevant with functional redundancy or similarity" is incorrect and unclear). If the authors had the opportunity to have the text reviewed by an English speaker, that would help. Alternatively, using a tool such as deepl.com could also greatly improve the clarity of the text.

Our comment: We again thank the reviewer for their critical comments, especially those relating to the cell-cycle arrest experiments. As the reviewer recommended, we had the manuscript proofread by a native English speaking science editor.

The work is quite descriptive, as mentioned by the authors in their response to reviewer 2, and provide an exhaustive description of the effects of remodeling factors at a relatively large scale. The authors should mention that using finer techniques to map contact changes, such as microC, may bring up additional details at the gene-scale level.

Our comment: We apologize for failing to mention MicroC-XL in the previous version of the manuscript. We added this information to the Discussion section (page 14, lines 17-20) of the revised manuscript.

I regret that the authors were not able to run a loop calling detection tool (chromosight or else, but that should be cited in the text) with more success, and I dont really understand their response to the original suggestion: "Since the signal of the yeast chromatin loops is weaker in the interphase (even in G2/M) compare to actual mitotic phase,..." Sup figure 5a is not really convincing, and I am sure that the analysis could have been done with much better results would the author have played with the program a bit. There are ways to quantify the loops strenghts and changes over the entire genome, to show the differences (or not) between IAA treated or not treated condition for each mutant, etc. Here the analysis looks preliminary and superficial, and i am not sure it really brings much to the text.

Our comment: A very recent paper by Koshland and colleagues (Constantino *et. al.*, 2020) found that dots (representing loops) appeared very weakly in the matrix of yeast Hi-C data.

Given that a weak loop signal cannot be clearly distinguished from the surrounding matrix in yeast, typical loop calling with tools such as 'Chromosight' would be very difficult. Therefore, we were unable to detect every loop.

The number of loops detected in our yeast data is very low (n=266), but this value is comparable to those obtained for yeast in the original article describing the 'Chromosight' method (Matthey-Doret *et al.*, 2020). We further processed these data and present the results in Supplementary Figure 8b-d. We also quantified the strength and other contact changes of a representative loop-like position, and present these additional data in revised Figure 3.

The use of references is also sometimes a bit strange.

line 65: then become denser as mitosis progresses^{25,26}. Refs 25 and 26 refers to meiosis, not mitosis. The authors probably think about schalbetter et al. 2017, not 2019? lazar et al 2017 also shows this. Same for the "in a CTCF-independent manner in *S. cerevisiae*^{22,23,26-28}": 26 should be schalbetter 2017, no? Ref 27 and 28 are in mammals, and should be cited earlier in the sentence, I think, and not after mentioning yeast... etc. The authors should double check their citations.

Our comment: We appreciate the reviewer's correction. We updated some references, including those mentioned above, and double-checked all references to ensure that we are accurate when expressing our appreciation for previous findings.

fastq files must be made available (and not just .cool files), so far they are not.

Our comment: We have deposited the raw data files (including the cell cycle arrest dataset) to GEO, but are unable to share the raw data files without making them 'public state', which we do not want to do prior to publication. We asked the GEO and SRA teams if there was a way to share fastq files with reviewers but, unfortunately, they replied that there is no way to share raw data through the GEO system. Please find below the full text of these email responses from the GEO and SRA teams.

(GEO passes the raw data for sequencing submissions to SRA. At this time, SRA does not routinely support access to private data. However, you can contact the SRA team (sra@ncbi.nlm.nih.gov) to request a private link to the SRA submission metadata. Downloadable files give information about submission structure, sample metadata, number of bases loaded per run, etc. This, perhaps, will help satisfy reviewers that the appropriate data have been deposited.) – from GEO team

(Hello Hyelim Jo, Sorry, but we do not have an option to share fastq files. Feel free to contact us with any questions or concerns.) – from SRA team

Reviewer #2 (Remarks to the Author):

This revised version of the manuscript addressed one of my main concerns, which was the difficulty of interpreting changes due to remodeling depletion in asynchronous populations. Data on synchronized cells are now presented and have confirmed the importance of comparing synchronized populations. Indeed, the new set of data shows that the effect observed for Sth1 was mainly due to the accumulation of cells in G2 upon depletion of Sth1.

FACS are now provided for the different conditions, which is nice, but Western blots to monitor the depletion are not shown.

Our comment: We thank the reviewer for these comments. As requested, we performed Western blotting with synchronized cells to monitor chromatin remodeler depletion upon IAA treatment. We now present these data in revised Supplementary Figure S3.

The effects of depleting individual remodelers (assuming that the depletion was effective in each case) on synchronized populations are very small. This is not a problem per se but raises the question of the significance of these observations for which no statistics are provided. Furthermore, the description of the data is either vague (e.g. p5 “ino80 depletion also slightly increased interchromosomal contacts”) or exaggerated (e.g. p 5 “among them depletion of Chd1, Swr1 and Sth1p yielded dramatic changes in 3D genome organization” or p10 “These observations imply that chromatin remodeling activity plays a key role in modulation of 3D genome structure » when the effects are barely visible and never quantified).

Our comment: As all results generated under the asynchronous condition were moved to Supplementary Figure 1 in the previous revision process, the above-mentioned description on page 5 refers only to our overall observation at the screening level, without any additional explanation or statistical data.

To address this comment, however, we sought to include more statistical data in the main figures and toned down the descriptions throughout the manuscript. We also revised the manuscript to avoid vague words (e.g., slightly and dramatic) and use fold-change values instead.

In response to my comment about the poor signal-to-noise ratio (also raised by reviewer 1), the authors choose to show interaction matrices at a 1kb resolution which is not helping at all to solve this issue. Matrices are even more noisy and difficult to interpret.

Our comment: We acknowledge and appreciate this constructive criticism, but we are still hesitant to use signal-to-ratio normalized data generated via tools such as HiCPlus or DeepHiC (including the recommended tool) for the following reasons:

The interactions of the yeast 3D conformation are weaker than those of the mammalian 3D conformation. Since yeast does not show strong interaction domains, such as TADs or loops, and most of the available tools are best suited for mammalian Hi-C data, the interaction signals can be unclear or exaggerated. Most of the previously reported yeast Hi-C data also did not undergo other normalization except SCN or ICE (including the research article on GSE90902, which we used for Scc1p analysis).

None of the analysis proposed by reviewer 1 and 2 was applied in this revised version and none of the observation is validated by a different approach.

Our comment: We apologize for giving the impression that we had not responded fully to the reviewers' comments in the previous revision. The previous revised version of the manuscript included various additional analyses, including validation of the reproducibility of analyses performed using the SCC value, as applied using HiCRep (Supplementary Table 4); and detection of the loop position, as performed using Chromosight (Supplementary Fig. 7).

In this revised version, we set out to further validate and substantiate our findings using various methods. Examples of these analyses can be found in updated Figures 2d, 2l, and 3e-h.

In general, this new version is poorly written and is sometimes even difficult to understand.

The raw data may be of interest to the community and these results could be published as a resource but not as a research article, in the current state.

Our comment: In order to increase readability, we submitted the manuscript to be proofread by an expert in English language science editing.

In this manuscript, we offer a compendium of chromatin contact maps reflecting the situation under regulation by chromatin remodelers. We focus on fully describing what we observed in this work. Based on our findings, we are willing to proceed to further detailed mechanistic studies of each chromatin remodeler in the future. According to a recent paper (Constantino et. al., 2020), the signal-to-noise ratio can be improved by using MicroC-XL for yeast. This would make it possible for us to perform a detailed analysis at the gene-scale level, greatly improving our potential to identify the molecular mechanism through which each chromatin remodeler impacts 3D genome organization in yeast.

In the revised manuscript, we now address the relationship between the cohesin association region (CAR) and Swr1p (Figure 3) to suggest a direction for further study.

Reviewer #3 (Remarks to the Author):

The authors have adequately addressed our comments in the revision. We feel the manuscript can be accepted.

Thank you once again for your valuable comments, and we want to emphasize that your feedback has helped us improve our paper.

Let me conclude this letter by thanking you for your time in processing this manuscript and the reviewers for their many helpful comments. Please let me know if there is anything else we can do, and thank you again for considering the manuscript.

Sincerely,

Daeyoup Lee, PhD.

P.S. The following is a list of changes made to the revised manuscript:

- 1) To respond to specific comment 1 of Reviewer #1 and several comments of Reviewer #2, we modified the text to use clear and easy-to-understand sentences. Furthermore, the revised manuscript has been proofread by native English speakers.
- 2) To respond to a comment of Reviewer #1, we carefully double-checked the references.
- 3) To respond to a comment of Reviewer #2, we performed additional Western blot experiments with synchronized cells (Supplementary Figure S3).
- 4) To respond to a comment of Reviewer #2, we performed further data analysis and added quantification data that validate our findings.

REVIEWERS' COMMENTS

Reviewer #1 (Remarks to the Author):

The authors have reviewed accordingly their manuscript. I am satisfied for the changes they brought to the newly revised version, and recommend publication.

Reviewer #2 (Remarks to the Author):

This new version of the manuscripts addresses several of the points that were raised in the first round of review. However, I still have some major concerns regarding some conclusions that are not supported by the data.

It will be too long to list them all, I will thus focus on the conclusions regarding the role of RSC in centromere clustering. Although in the result section the authors acknowledge the marginal effect of Sth1 ("Sth1p depletion, which caused marginal changes throughout the cell cycle" and later "The inter CEN-CEN interactions also increased upon depletion of Sth1p, but this change was marginal compared to that seen upon Scc1p depletion") they claim in the abstract that: "RSC appeared to support the function of cohesin in centromeric clustering at G2 phase". The effect of Sth1 (if any) is indeed marginal (see figure 4f lower panel) and, from my point of view, it would not disserve to be mentioned in the abstract. Furthermore, I don't see any experiment supporting the conclusion that Sth1 facilitates the function of cohesin.

Along the same line: line 301 it is written "These findings suggest that the cohesin, Scc1p, is sufficient to manage inter CEN-CEN interactions at G2 phase, and that Sth1p may facilitate this function of Scc1p."

It is not clear what the authors mean by sufficient here. I don't understand why the authors insist so much on the effect of Sth1 on centromere clustering while the effect is barely visible. Again, I don't see any experiment showing that Sth1 affect centromere clustering through Scc1.

The other effects shown for the depletion of Sth1 are not quantified and are either anecdotal and/or barely visible. In particular, there is no quantification for the (very weak) effects of Sth1 depletion or Sth1 catalytic mutant shown in panels 4a-d. Panel 4h: Again, it is difficult to evaluate the robustness of this result. Furthermore, it is not clear whether the 0.37-0.39 Mb region of chromosome 5 is the only locus "affected" by Scc1 and Sth1 and how many other loci are "affected" by each of these factors? Assuming that Sth1 and Scc1 are regulating local interaction at this locus: What is specific about this locus? What are the genes/regulatory sequences involved? Is Scc1 enriched in this region?

In the discussion the authors conclude "Notably, Rab1 configurations are actively maintained by the cohesin and RSC complexes at centromeric loci, indicating that specific roles are played by the cohesin-mediated chromatin folding mechanism governed by the RSC complex"

This is again an overstatement (see above for the impact of Sth1 depletion and the relationship with cohesin). Furthermore, the Rab1 configuration refers not only to centromere clustering (which is not abolished upon Scc1 depletion - but only reduced 1.6 fold - and barely affected by Sth1 depletion) but also to the subnuclear position of centromeres and telomeres (close to the nuclear periphery on opposite sides of the nucleus), which is not assessed here.

Finally, new data are provided in this revised version, in particular the western blot showing the extent of auxin-induced degradation for each factor in each condition (supplementary figure 3). These western blots clearly show that the amount of some factors is only slightly reduced in some conditions; this should have been mentioned and discussed in the main text. For instance: Sth1 shows a modest reduction in G2 synchronized cells (figure S3C), possibly accounting for the weak effect that they observed on chromosome conformation in this condition.

Figures:

Figures have been improved in this new version of the manuscript. In particular, the Figure 2, it would be nice to show the same representation as panel 2d for the depletion of Chd1 and Sth1 (comparing the Log₂ ratio of the average contact probability along the genomic distance between control and IAA-treated (+IAA) samples in G1, S and G2 phases).

Although the English have been revised, there are still some enigmatic sentences remaining
Line 341: "Indeed, Sth1p showed diverse 3D genome-organizing functions depending on the chromosomal context and/or cell cycle stage, even though all chromatin remodelers share common biochemical activities (e.g., the ability to alter histone-DNA interactions)." I don't understand the link between the two parts of this sentence: why mentioning other remodelers when discussing the diverse role of Sth1 depending on the chromosomal context?

We sincerely appreciate the reviewer's comments. Please find below our detailed response to the reviewers' comments.

Reviewer #1 (Remarks to the Author):

The authors have reviewed accordingly their manuscript. I am satisfied for the changes they brought to the newly revised version, and recommend publication.

Our comment: We thank the reviewer #1 for the positive evaluation of our work.

Reviewer #2 (Remarks to the Author):

This new version of the manuscripts addresses several of the points that were raised in the first round of review. However, I still have some major concerns regarding some conclusions that are not supported by the data.

It will be too long to list them all, I will thus focus on the conclusions regarding the role of RSC in centromere clustering. Although in the result section the authors acknowledge the marginal effect of Sth1 ("Sth1p depletion, which caused marginal changes throughout the cell cycle" and later "The inter CEN-CEN interactions also increased upon depletion of Sth1p, but this change was marginal compared to that seen upon Sccl1p depletion") they claim in the abstract that: "RSC appeared to support the function of cohesin in centromeric clustering at G2 phase". The effect of Sth1 (if any) is indeed marginal (see figure 4f lower panel) and, from my point of view, it would not deserve to be mentioned in the abstract. Furthermore, I don't see any experiment supporting the conclusion that Sth1 facilitates the function of cohesin.

Along the same line: line 301 it is written "These findings suggest that the cohesin, Sccl1p, is sufficient to manage inter CEN-CEN interactions at G2 phase, and that Sth1p may facilitate this function of Sccl1p."

It is not clear what the authors mean by sufficient here. I don't understand why the authors insist so much on the effect of Sth1 on centromere clustering while the effect is barely visible. Again, I don't see any experiment showing that Sth1 affect centromere clustering through Sccl1. The other effects shown for the depletion of Sth1 are not quantified and are either anecdotal and/or barely visible. In particular, there is no quantification for the (very weak) effects of Sth1 depletion or Sth1 catalytic mutant shown in panels 4a-d. Panel 4h: Again, it is difficult to evaluate the robustness of this result. Furthermore, it is not clear whether the 0.37-0.39 Mb region of chromosome 5 is the only locus "affected" by Sccl1 and Sth1 and how many other loci are "affected" by each of these factors? Assuming that Sth1 and Sccl1 are regulating local interaction at this locus: What is specific about this locus? What are the genes/regulatory sequences involved? Is Sccl1 enriched in this region?

Our comment: We focus on this point (Sth1p and Sccl1p) because several previous studies have mainly focused on the interaction between cohesin complex and RSC (Muñoz et al., 2019, Lopez-Serra et al., 2014, Huang et al., 2004; *these papers have been properly cited in the manuscript.*).

The previous studies have reported that RSC and cohesin complex interact with each other at centromere and chromosomal arm regions. These previous studies also suggest that RSC mediates cohesin function and localization on chromatin. Furthermore, it is well known that Sth1p localizes at the centromeric region and regulates the centromeric structure and sister

chromatid cohesion (Lopez-Serra et al., 2014, Hsu JM et al., 2003, Tsuchiya et al., 1998).

Previous studies on both of Sth1p and Scc1p proteins have been mainly carried out using the following experimental methods: 1) The effect of RSC on cohesin localization was investigated by using ChIP and MNase-seq. 2) Sister chromatid cohesion defect was observed under a microscope. 3) The physical interaction between the two proteins was confirmed by *in vitro* experiments. 4) The regulatory effect on the centromere structure was observed through *in vitro* experiments including chromosome segregation assay and chromatin digestion assay.

We thought this topic (about RSC and cohesin) was worth revisiting again and wanted to investigate two proteins actually interact in regulating the 3D structure of chromatin through the *in situ* Hi-C. Surprisingly, our work yielded some inconsistent results with the previous studies. The 3D structural collapse patterns induced by Sth1p and Scc1p deficiency were different.

However, we do not want to completely deny other group's works for the following reasons:

1) Our data partially support the results of previous studies.

: Our *in situ* Hi-C data suggest that the Sth1p contribute the centromere clustering. As you mentioned, the effect of Sth1p on centromere clustering "is barely visible" compare to Scc1 in Fig 4f. However, a point centromeric region (centromere +/-5kb region) was significantly collapsed upon *sth1p*-depletion in Fig 4g (p-value was 0.001). It implies that Sth1p is involved in centromere clustering at this point centromeric region. The reason that the change upon Scc1p-depletion is clearly visible on the heatmap (Fig 4f) may be because Scc1 acts in a wider range (centromere +/-50kb region) than Sth1.

2) Our findings are inconsistent with the results of previous studies but it does not mean that the previous works were wrong.

: As we mentioned this issue in discussion, the overall mechanism of 3D genome organization and the relationship between chromosomal 3D architecture and gene expression/ protein localization/ nucleosomal structure are not fully understood yet.

Therefore, we decided to provide an inclusive interpretation to reflect previous findings.

It is well known that Scc1p mainly regulate the 3D genome organization including centromeric architecture and the Sth1p mediates the Scc1p localization at least in the centromere flanking region. Thus, we speculated that the Sth1p may support the regulatory function on centromere clustering of the Scc1p.

In the discussion the authors conclude "Notably, Rab1 configurations are actively maintained by the cohesin and RSC complexes at centromeric loci, indicating that specific roles are played by the cohesin-mediated chromatin folding mechanism governed by the RSC complex" This is again an overstatement (see above for the impact of Sth1 depletion and the relationship with cohesin). Furthermore, the Rab1 configuration refers not only to centromere clustering (which is not abolished upon Scc1 depletion - but only reduced 1.6 fold - and barely affected by Sth1 depletion) but also to the subnuclear position of centromeres and telomeres (close to

the nuclear periphery on opposite sides of the nucleus), which is not assessed here.

Our comment: Thank you for your critical comments. Considering that the mentioned part did not sufficiently reflect the revised data, the part was deleted. However, we still argue that the chromatin remodeling can influence the Rab1 configuration.

The Rab1 configuration refers not only to centromere clustering but centromere clustering is one of the important components of Rab1 configuration and the condensation or de-condensation of chromosomal arms also affects the overall 3D configuration.

Our result did not demonstrate 3D structural changes in every part of Rab1 configuration (e.g., telomere, rDNA locus) but clearly show the changes in centromere clustering and also chromosomal arm condensation. The condensation or de-condensation of chromosomal arms also affects the overall 3D configuration.

Finally, new data are provided in this revised version, in particular the western blot showing the extent of auxin-induced degradation for each factor in each condition (supplementary figure 3). These western blots clearly show that the amount of some factors is only slightly reduced in some conditions; this should have been mentioned and discussed in the main text. For instance: Sth1 shows a modest reduction in G2 synchronized cells (figure S3C), possibly accounting for the weak effect that they observed on chromosome conformation in this condition.

Our comment: It is an important point to consider, but it cannot be assumed that the quantitative amount of a factor is precisely proportional to their capacity in 3D genome organization.

The below table shows a quantified value of western blot (Supplementary Figure 3):

CHD1						
	G1_no	G1_IAA	S_no	S_IAA	G2_no	G2_IAA
A.Target_Signal	96.695	55.768	77.299	43.995	64.632	44.875
B.Tubulin_Control	150.504	176.807	185.449	179.732	169.712	155.655
Relative_Signal (A/B)	0.6424746	0.3154174	0.4168208	0.2447811	0.3808334	0.2882978
Reduction	0.50905862		0.41274255		0.242981763	

SWR1						
	G1_no	G1_IAA	S_no	S_IAA	G2_no	G2_IAA
A.Target_Signal	60.714	38.734	66.755	37.243	80.63	36.319
B.Tubulin_Control	96.613	97.961	105.935	106.945	102.254	119.126
Relative_Signal (A/B)	0.6284247	0.3954023	0.6301506	0.3482444	0.7885266	0.3048789
Reduction	0.370804135		0.447363147		0.613356273	

STH1						
	G1_no	G1_IAA	S_no	S_IAA	G2_no	G2_IAA
A.Target_Signal	90.082	35.85	52.331	35.509	50.322	40.785
B.Tubulin_Control	122.596	120.008	125.982	118.556	107.82	134.23
Relative_Signal (A/B)	0.7347874	0.2987301	0.4153847	0.2995125	0.4667223	0.3038441
Reduction	0.593446932		0.278951681		0.348983028	

As you can see the amount of protein varies depending on cell cycle stage. Chd1p and Sth1p

was most abundant in the G1 state while Swr1p was most abundant in the G2 state. Similarly, Chd1p depletion causes largest change in 3D architecture of chromosome in G1 (up to 1.5-fold changes in contact probability). Swrp1p depletion causes largest change in G2 phase.

Considering only this observation, as you mentioned, it can be assumed that there is a correlation between the reduction efficiency of target protein and the degree of changes in 3D structure. Also, it can be assumed that the most dramatic changes are observed when the target protein is deficient in the most resident cell cycle.

However, in the case of Sth1p, although the reduction rate of Sth1p in the G1 phase (0.59) was twice that of the S phase (0.28), there was no significant difference in the 3D structure change between the G1 and S phases. In addition, the Sth1p (0.59) was similarly depleted compared to Chd1p (0.51) in the G1 phase, but the 3D structural change upon Sth1p-depletion was insignificant compared to that of Chd1p.

Figures:

Figures have been improved in this new version of the manuscript. In particular, the Figure 2, it would be nice to show the same representation as panel 2d for the depletion of Chd1 and Sth1 (comparing the Log2 ratio of the average contact probability along the genomic distance between control and IAA-treated (+IAA) samples in G1, S and G2 phases).

Our comment: In response to this comment, we updated supplementary Figure 4. We added figure for Swr1p(d) and Sth1p(e) with the same representation as Figure 2d.

Although the English have been revised, there are still some enigmatic sentences remaining Line 341: "Indeed, Sth1p showed diverse 3D genome-organizing functions depending on the chromosomal context and/or cell cycle stage, even though all chromatin remodelers share common biochemical activities (e.g., the ability to alter histone-DNA interactions)." I don't understand the link between the two parts of this sentence: why mentioning other remodelers when discussing the diverse role of Sth1 depending on the chromosomal context?

Our comment: In response to this comment, we modified the sentences in the discussion section (Page 14 line 3-4 and 17-19).

Let me conclude this letter by thanking you for your time for their many helpful comments.

Sincerely,

Daeyoup